

# Quantification of Methane Sources in the Athabasca Oil Sands Region of Alberta by Aircraft Mass-Balance

Sabour Baray[1], Andrea Darlington[2], Mark Gordon[3], Katherine L. Hayden[2], Amy Leithead[2], Shao-Meng Li[2], Peter S.K. Liu[2], Richard L. Mittermeier[2], Samar G. Moussa[2], Jason O'Brien[2], Ralph Staebler[2], Mengistu Wolde[4], Doug Worthy[5], Robert McLaren[1]

1. Centre for Atmospheric Chemistry, York University, Toronto 2. Air Quality Research Division, Environment and Climate Change Canada, Toronto 3. Earth and Space Science and Engineering, York University, Toronto 4. National Research Council of Canada, Ottawa 5. Climate Research Division, Environment and Climate Change Canada, Toronto, Canada

Corresponding authors: Robert McLaren (rmclaren@yorku.ca) & Katherine Hayden (katherine.hayden@canada.ca)

**Abstract.** Aircraft-based measurements of methane ($CH_4$) and other air pollutants in the Athabasca Oil Sands Region (AOSR) were made during a summer intensive field campaign between August 13 and September 7 2013, in support of the Joint Canada–Alberta Implementation Plan for Oil Sands Monitoring. Chemical signatures were used to identify $CH_4$ sources from tailings ponds (BTEX VOC's), open-pit surface mines ($NO_y$ and rBC) and elevated plumes from bitumen upgrading facilities ($SO_2$ and $NO_y$). Emission rates of $CH_4$ were determined for the five primary surface mining facilities in the region using two mass balance methods. Emission rates from source categories within each facility were estimated when plumes from the sources were spatially separable. Tailings ponds accounted for 45% of total $CH_4$ emissions measured from the major surface mining facilities in the region while emissions from operations in the open pit mines accounted for ~50%. The average open pit surface mining emission rates ranged from 1.2 to 2.8 tonnes of $CH_4$ hr$^{-1}$ for different facilities in the AOSR. Amongst the 19 tailings ponds, Mildred Lake Settling Basin, the oldest pond in the region, was found to be responsible for the majority of tailings ponds emissions of $CH_4$ (>70%). The sum of measured emission rates of $CH_4$ from the five major facilities, $19.2 \pm 1.1$ tonnes $CH_4$ hr$^{-1}$, was similar to a single mass balance determination of $CH_4$ from all major sources in the AOSR determined from a single flight downwind of the facilities, $23.7 \pm 3.7$ tonnes $CH_4$ hr$^{-1}$. The measured hourly $CH_4$ emission rate from all facilities in the AOSR is $48 \pm 8\%$ higher than that extracted for 2013 from the Canadian Green House Gas Reporting Program, a legislated facility-reported Emissions Inventory, converted to hourly units. The measured emissions correspond to an emissions rate of $0.17 \pm 0.01$ Tg $CH_4$ yr$^{-1}$, if the emissions are



assumed temporally constant, an uncertain assumption. The emission rates reported here are relevant for the summer season. In future, effort should be devoted to measurements in different seasons to further our understanding of seasonal parameters impacting fugitive emissions of $CH_4$ and to allow better estimates of annual emissions and year to year variability.



## Introduction

Methane ($CH_4$) is a significant greenhouse gas (GHG), second in rank to carbon dioxide ($CO_2$) in terms of its direct radiative forcing (Montzka et al., 2011;IPCC, 2013). Sources of $CH_4$ to the atmosphere fall into three broad categories: biogenic (animal husbandry, waste/landfills, wetlands, agriculture),

pyrogenic (biomass burning) and thermogenic (coal bed methane, fossil fuel reservoirs, etc).  In order of importance, the sinks of $CH_4$ include reaction with the hydroxyl radical (OH) in the troposphere (Vaghjiani and Ravishankara, 1991), destruction in the stratosphere, soil uptake processes (IPCC, 2013) and reaction with Cl atoms in the upper troposphere-lower stratosphere.  With an atmospheric lifetime of about 9 years, $CH_4$ is first oxidized to formaldehyde (HCHO), which is subsequently photolyzed or

oxidized to yield CO and eventually $CO_2$.  Controlling emissions of $CH_4$ is an attractive climate control strategy because of its short lifetime and its larger global warming potential (GWP) and global temperature change potential (GTP) compared to $CO_2$ (IPCC, 2013).

In addition to climate implications, $CH_4$ can also have air quality implications through its role in $NO_x$ catalysed ozone formation in the troposphere on global and regional scales.  Recently, in regions of

intense oil and gas extraction with advanced directional drilling and hydraulic fracturing, boundary layer $O_3$ greater than 140 ppb has been observed frequently in wintertime (Pinto, 2009;Schnell et al., 2009).  These high $O_3$ levels are associated with enhanced VOC levels (including $CH_4$) and moderate $NO_x$ levels that result from the oil and gas extraction activities.  These rapid ozone formation events (OFEs) in winter are interesting not only because snow coverage is a necessary ingredient (Edwards et

al., 2014), but also due to their contrast to both summertime smog events (Finlayson-Pitts and Pitts, 1997) and ozone depletion events (ODE's) over snow in arctic regions (Barrie et al., 1988).  Further study has shown that the winter OFE's are driven by stagnation in shallow boundary layers (Schnell et al., 2016), enhanced photolysis rates over snow, high levels of ozone precursors and photolysis of HCHO and other carbonyls under radical limited conditions during these events (Edwards et al., 2014).

The budget of HCHO during OFE's is still uncertain; however, oxidation of $CH_4$ at levels up to 10 times background likely plays a role in the budget that is not insignificant.



Following an increase in the atmospheric burden of $CH_4$ in the post industrial revolution, growth of $CH_4$ slowed in the 1980's and 90's, simultaneous with the economic collapse of the Soviet Union and a reduction of emissions from fossil fuel exploration in Eurasia (Worthy et al., 2009). The pause in the growth of global methane observed from ~1990-2007 was only temporary as evidenced by the recent

increase seen since 2007, with a current atmospheric growth rate of ~0.4% $yr^{-1}$ from 2007-2014 in the northern hemisphere (Hausmann et al., 2016). Presuming that the lifetime of methane was constant during this period, the imbalance suggests an additional net methane source of ~24-45 Tg $CH_4$ per year. Increased tropical precipitation and anomalously higher temperatures in the Arctic were first suggested as drivers for the increase in atmospheric $CH_4$ (Dlugokencky et al., 2009). More recently total increases

in tropospheric columns of ethane and methane have been used to deduce minimum contributions, C, of the global oil and gas sector to the post 2007 methane increase of C >39%, using realistic ethane to methane emission ratios (Hausmann et al., 2016). Satellite observations have been used to suggest an increase in $CH_4$ emissions in the USA of 30% over the 2002-2014 period (Turner et al., 2016), while a similar 2009-2014 trend in global mixing ratios and tropospheric columns of ethane is also attributed to

oil/gas production in the USA (Helmig et al., 2016). On the other hand, a recent study suggests that North American $CH_4$ emissions have been flat over the years 2000 -2012 (Bruhwiler et al., 2017) and there is still ambiguity in the source versus sink role in the recent increase in atmospheric $CH_4$ (Turner et al., 2017). Clearly there are still uncertainties in the contribution of anthropogenic emissions of $CH_4$ to the atmosphere and emissions of CH4 need further quantification.

Emission inventories help determine the contributions of specific sources to trace gas levels in the atmosphere. However, bottom-up emission inventories benefit from top-down measurements and validation (Fujita et al., 1992), due to the difficulty in identifying all possible points of emission and accurately calculating emissions from all those points in a large complex source (e.g., a city or a facility). Top-down measurements of various types have long been used in the validation of emission

inventories and emission factor models including comparison of surface based pollutant profiles and ratios (Fujita et al., 1992;Jiang et al., 1997;Fujita et al., 1995), source-receptor methods (Scheff and Wadden, 1993;Fujita et al., 1995;McLaren et al., 1996), aircraft based flux measurements (Mays et al., 2009), measurement/modelling hybrid methods (Allen et al., 2004;Shephard et al., 2015) and satellite



measurements (McLinden et al., 2012;McLinden et al., 2014;Turner et al., 2016;Kort et al., 2014;Jacob et al., 2016). Multiple studies have suggested underestimation of $CH_4$ emissions from natural gas infrastructure (Brandt et al., 2014;Hendrick et al., 2016). Several recent aircraft studies using mass balance approaches have quantified $CH_4$ emissions in oil and gas regions and compared these to

emissions inventory emission rates and/or leakage rates (Karion et al., 2013;Peischl et al., 2013;Karion et al., 2015;Peischl et al., 2015;Peischl et al., 2016;Lavoie et al., 2015).  Other studies have used top-down satellite measurements to quantify emission of $CH_4$ in oil and gas regions (Schneising et al., 2014;Kort et al., 2014). As such, top down measurement of methane emissions and comparison with bottom up inventories can make a significant contribution to our understanding of the global budget of

$CH_4$.

In this study we quantify total emission rates of $CH_4$ from facilities in the Athabasca Oil Sands Region (AOSR) of Alberta in the summer of 2013.  Alberta has large deposits of oil sands, an unconventional viscous mixture of bitumen, sand, silt, clay, water and trapped gases (Stringham, 2012).  Canada has proven reserves of $1.69 \times 10^{11}$ barrels of oil ($2.7 \times 10^{13}$ litres), third largest in the world, 97% of which are

located in the oil sands [Orbach, 2012].  Approximately 82% of the oil sands are located in the AOSR north of Fort McMurray with 20% located in near surface deposits (depth < 100m) that can be mined using open pit techniques and the remainder located in deeper deposits requiring underground in-situ extraction.  In both cases the oil must be separated from sand requiring the use of hot water or steam froth treatment, and organic solvent diluents (naphtha or paraffin) are used to help separate water and

solids and/or to decrease the bitumen viscosity. For surface mining processes, once the bitumen is separated, process water containing unrecovered organic diluents is recycled but some is discharged in large tailings ponds open to the atmosphere for further remediation. Oil extraction in the AOSR is unique in that unlike other oil and gas regions, $CH_4$ is not the primary economic commodity being extracted, but is an unintended by-product.  In particular, a significant fraction of the $CH_4$ is not

associated with fossil fuel reserves, but is emitted from tailings ponds (Small et al., 2015).  Additional fugitive $CH_4$ is associated with the gaseous component of the oil sand along with other gases (Strausz, 2003;Johnson et al., 2016) that are released during overburden removal, open pit mining and/or subsequent processing.





In the summer of 2013, an intensive ambient air measurement campaign took place in the AOSR with both ground and airborne components in support of the Joint Oil Sands Monitoring (JOSM) Plan (JOSM, 2012). The airborne measurements were conducted to address four objectives: i) to measure and quantify air emissions from the oil sands mining facilities, ii) to study the downwind physical and

chemical transformation of pollutants emitted, iii) to provide spatio-temporal measurement of pollutants suitable for intercomparison with simultaneous satellite nadir overpasses in the region, and iv) to support air quality model prediction capabilities. In this paper, we report $CH_4$ emissions from industrial facilities in the AOSR based on the airborne campaign.  We applied the Top-down Emissions Rate Retrieval Algorithm (TERRA) mass-balance approach (Gordon et al., 2015) to determine total $CH_4$

emissions rates from each of the major industrial facilities, as well as a second mass-balance approach using downwind flight tracks to spatially separate $CH_4$ emissions from different sources in each facility. Emissions rates of $CH_4$ are determined for the five major facilities in the region: Syncrude Mildred Lake (SML), Suncor Energy OSG (SUN), Canadian Natural Resources Limited Horizon (CNRL), Shell Albian Muskeg River and Jackpine (SAJ) and Syncrude Aurora (SAU). These results are the first

source-attributed emissions estimates for the facilities in the AOSR, obtained by identifying and characterizing plume origins according to signatures of chemical tracer species.

**Experimental**

**2.1 Instrumentation**

An array of instruments for the measurement of trace gases, aerosols, meteorological and aircraft state
parameters were installed aboard the National Research Council of Canada Convair 580 research aircraft. Measurements of $CH_4$, $CO_2$, CO and $H_2O$ were made using a cavity ring-down spectrometer (Picarro G2401-m) at an interpolated rate of approximately 0.5 Hz with a flow rate of ~435 sccm min$^{-1}$. The precision of the $CH_4$ measurement was 2 ppb, and the uncertainty of the measurement at background (~ 1.9 ppm) was 3.3 ppb (@2 sec).  The instrument was calibrated six times before, during
and after the project using two standard reference gases traceable to NOAA GMD standards.  Methane mixing ratios are reported throughout as dry mole fractions in the paper. Necessary parameters for emissions estimation included Temperature (T), measured using Rosemount probe, Dewpoint



temperature ($T_d$), measured with an Edgetech hygrometer, and pressure (P), measured with a DigiQuartz sensor. The three-component wind speed (Ux, Uy, Uz) was derived from a Rosemount 858 probe, GPS and Honeywell HG1700 inertial measurement unit. The uncertainty of horizontal and vertical winds on the aircraft are 0.6 and 0.4 m/s respectively (Williams and Marcotte, 2000).

Geospatial information (latitude, longitude, ellipsoid height altitude) was measured by GPS.

Nitrogen oxides (NO, $NO_2$ and $NO_y$) were measured with a modified trace level chemiluminescent analyser (Thermo Scientific Model 42i-TL). A molybdenum converter (325 °C) was used to convert $NO_y$ species to NO and an $NO_2$ specific converter (Droplet Measurement Technologies) was used to convert $NO_2$ to NO. Detection limits for NO, $NO_2$, and $NO_y$ were determined to be 0.08 ppb (1 sec),

0.20 ppb (2 sec), 0.09 ppb (1 sec) respectively. Sulfur dioxide ($SO_2$) was measured with a pulsed UV fluorescence analyser (Thermo Scientific: Model 43i-TLE) with a detection limit of 0.7 ppb (1 sec). Ambient air was drawn in through filtered 6.35 mm (1/4") diameter PFA tubing taken from a rear-facing inlet located on the roof toward the rear of the aircraft. Measurements of NO, $NO_2$, $NO_y$ and $SO_2$ were made downstream of a constant pressure inlet system maintained at 770 mmHg with a total flow

rate of 5 Lpm. In flight zero and background determinations were made several times throughout each flight and the analysers were calibrated multiple times during the study against National Institute and Standards Technology (NIST) certified reference gases.

Refractory black carbon mass (rBC), was measured with a Droplet Measurement Technologies (DMT) Single Particle Soot Photometer (SP2). Ambient air was subsampled from the main aerosol flow that

was brought into the main cabin with a forward-facing shrouded diffuser isokinetic aerosol inlet (Cheng et al., 2017). Benzene, toluene, ethylbenzene and xylenes (BTEX) were measured by a Proton Transfer Reaction time-of-flight Mass Spectrometer (PTRMS) from the main gas inlet. Further technical details are provided elsewhere (Li et al., 2017). The delay time of each instrument was determined experimentally and through calculations based on sample flow rates and inlet volumes. Total delays are

contributed to by the response time of the instruments (1–3 sec) as well as the volume of sampling tubing. Data were adjusted to account for the total delay times of 2-6 sec to spatially and temporally synchronize the different measurements (Picarro delay time = 6 sec). The average speed of the aircraft



was 90 ms⁻¹ during the research flights, thus providing a spatial resolution of 90-270 m based upon the internal response time of each measurement.

## 2.2 Aircraft Flights

In total, there were 22 flights with 84 hours of measurements in the AOSR between August 13 and
September 7, 2013. The flights were designed for three purposes; measurement of pollutant emissions from facilities (Gordon et al., 2015;Li et al., 2017), measurement of pollutant transformation downwind of the AOSR (Liggio et al., 2016) and comparison with satellite overpasses (Shephard et al., 2015). Thirteen flights were dedicated to quantifying facility emissions with a minimum of two flights for each of the SML, SUN, CNRL, SAJ and SAU facilities. $CH_4$ above background was not detected during the
2013 flights targeting the Imperial Kearl Lake (IKL) facility, which was not in full production mode at the time (but has since expanded significantly), nor from the Suncor Firebag in-situ operation. We did detect $CH_4$ above background suspected to originate from the Suncor MacKay River operation (west of SML). We were not able to quantify this source separately, however emissions from this source are included in one measurement of the total emissions from all mining facilities in the AOSR using a wide
downwind screen (see section 3.4 and Fig. 7). Several other flights are not included in the analysis due to unfavourable meteorological conditions including wind shear problems or insufficient numbers of transects. In total, seven flights were found to be suitable for identifying and quantifying emissions of $CH_4$ from the facilities. Figure 1 displays several of the flight tracks over and downwind of the target facilities north of the Fort McMurray airport.

The flight patterns designed for the quantification of emissions rates were of two types: i) screen flights, wherein the aircraft flew transects perpendicular to the plume downwind of one or more facilities, and ii) box flights, wherein the aircraft flew transects at multiple heights around a single target facility in a box-type pattern (Gordon et al., 2015;Li et al., 2017). The transects were performed at heights from 150 to 1370 m above ground level (agl), complemented by vertical profiles designed to determine the height
of the planetary boundary layer (PBL) and to compare with ground based measurements.



## 2.3 Mass Balance Approaches for Determining CH₄ Emissions

Following the TERRA methodology (Gordon et al., 2015), the time resolved measurements were interpolated using covariance kriging to produce a 40 m (horizontal, s) by 20 m (vertical, z) contiguous screen of $CH_4$ mixing ratios. Within TERRA, the $CH_4$ mixing ratios are extrapolated from the lowest

transect (~150 m agl) to the surface using a constant, linear or half-gaussian extrapolation, depending on the type of source and the boundary layer conditions at the time. Uncertainty estimates (see section in Supplemental) are included according to the various types of surface extrapolation applied. Interpolated matrices were constructed for measurements of air pressure ($P_{air}$) and temperature ($T_{air}$) in order to determine the air mass balance within the box and to convert mixing ratios to mass densities. Spatially

equivalent interpolations of wind velocity perpendicular to aircraft motion ($U_\square$) were created from the vector components of wind speed and direction measurements.

Emissions rates were determined according to the two different mass-balance approaches for screen and/or box flight patterns. Horizontal tracks at multiple altitudes flown perpendicular to the general wind direction produce a virtual screen downwind of the target that is intercepted by emission plumes

from the facilities. Fluxes of $CH_4$ moving through each 40 x 20 m (s × z) pixel can be determined from the interpolated matrices and integrated for a dimensional s by z target area according to Equation 1

$$E_{Screen} = \iint_{s1,z_b}^{sn,z_t}([CH_4] - [CH_4]_B) \times U_\square \, ds \, dz \qquad \text{(Eq. 1)}$$

where ($[CH_4] - [CH_4]_B$) is the enhanced mixing ratio of $CH_4$ above background, $U_\square$ is the horizontal wind velocity perpendicular to the screen (e.g. $U \times \sin\theta$, $\theta$ = angle between wind vector and airplane vector), $s_1$ and $s_n$ are the horizontal integration limits along the screen transect, $z_b$ and $z_t$ are the bottom and top vertical integration limits. Background mixing ratios of $CH_4$ were determined from the outside edges of the screens away from plume sources. Because $[CH_4]_B$ varies with height, a vertically variant

background profile was subtracted from each vertical measurement column, an approach used in other mass balance determinations (Cambaliza et al., 2014;Karion et al., 2013). The simple mass balance



approach represented by Eq.1 can be applied to individual downwind screens from other flight paths (i.e. box flights) to determine $CH_4$ emissions from specific sources within a facility.

The second mass balance method used in this paper is to apply the full box-model TERRA algorithm (Gordon et al., 2015), to compute total emissions from all sources within a box, where the box is made up of multiple (4-6) screens forming a polygon encompassing a facility. This more rigorous mass balance approach used for calculating total emissions from a facility is represented by Equation 2

$$E_{Box} = E_{CH} + E_{CHT} + E_{CV} + E_{CVT} + E_{CM} \qquad \text{(Eq. 2)}$$

where $E_{Box}$, the total emissions rate from all sources within a box, is the sum of the horizontal advective and turbulent fluxes ($E_{CH}$ and $E_{CHT}$), vertical advective and turbulent fluxes ($E_{CV}$ and $E_{CVT}$), and the change in $CH_4$ mass within the box volume ($E_{CM}$). Because the box includes screens that are downwind, upwind and lateral to sources, incoming (background) and outgoing (background + source) fluxes are determined as a part of the horizontal flux terms ($E_{CH}$ and $E_{CHT}$). Vertical fluxes through the box top, normally ignored in the conventional mass balance approaches (Eq. 1), are estimated according to the conservation of air mass within the box volume and the mixing ratio at the top edge of the box. $E_{CM}$ is estimated according to the time derivative of the ideal gas law, based on measured changes in pressure and temperature over the flight time (see Gordon et al., 2015 for a full discussion).

The advantage of the box approach (Eq. 2) over the screen approach is a more precise estimate of total emissions by accounting for incoming and outgoing fluxes and meteorological effects within a volume. However this flight pattern takes more time to completely surround a target facility. The advantage of the screen approach (Eq. 1) is the computation of $CH_4$ fluxes per pixel, which can thus be used to spatially integrate individual emission plumes of arbitrary shapes when the sources can be spatially resolved. Studies applying aircraft mass-balance methods have used each of single-height transect (Karion et al., 2013;Peischl et al., 2016), single screens (Cambaliza et al., 2014;Walter et al., 2012), spiral (Wratt et al., 2001;Gatti et al., 2014), and full box flight paths (Gordon et al., 2015) for the



purpose of determining emissions rates and characterizing meteorological conditions. The aircraft flights presented contained various segments of tracks that allowed applications of all the above methods. In this work we apply a systematic approach deriving information from each of these techniques for a comprehensive top-down characterization of $CH_4$ sources and emissions in the region.

Single-height transects are used to determine source chemical signatures by identifying $CH_4$ enhancements and their associations with other trace-gas species. Vertical profiles are used to determine the PBL height throughout flights. Single screens are used to determine $CH_4$ emissions rates (Eq. 1) for facilities and their individual sources when plumes are spatially resolved. Box flights are used to determine total $CH_4$ emissions from facilities at a lower uncertainty (Eq. 2) and source-specific

emissions are determined where possible (Eq. 1).

The summer time emission rates measured in this study are reported in units of metric tonnes $CH_4$ per hour, an appropriate unit given the duration of the flights (i.e., a few hours). We do not attempt to derive annual emissions as the assumptions needed to do so are highly uncertain without measurements

in other seasons for a volatile species such as $CH_4$. However, we do make a first order comparison to emission inventories and other studies that report emissions on an annual basis by downscaling the annual emissions to hourly emission rates using an assumption of a constant temporal factor throughout the entire year. This is appropriate for emission inventories that are based upon measurement of emissions or emission factors in summer, that then upscale their emission rates of $CH_4$ to annual

emissions using a constant temporal factor assumption (e.g., GOA, 2014). However the assumption of a constant temporal factor is far from being validated and further measurements in different months are needed to understand the potential for seasonal variability of fugitive emissions of $CH_4$.

Previous work shows the box approach has a demonstrated uncertainty of 25-27% for total emissions of

$CH_4$ from a facility in the AOSR (Gordon et al., 2015). Uncertainty due to extrapolation of $CH_4$ mixing ratios from the lowest height measurements to the surface was estimated to be 15% and 26% in that study. In contrast, screen approaches used in other studies have estimated uncertainty in the range of 30-50% (Cambaliza et al., 2014) with the main sources of error attributed to the reliability of plume



characterization and the stability of meteorological conditions. In this study, uncertainties in both the box and screen estimates are reduced through i) a high number of transects over a wide vertical range to accurately characterize vertical structure in the PBL, ii) reliable measurements of background $CH_4$ (or incoming fluxes for boxes), iii) measurements of the PBL height to account for meteorological variance,

and iv) measurements within time periods of minimal PBL change. In addition, the enhancement of $CH_4$ in the plumes downwind of the facilities and high precision of the Picarro instrument minimize uncertainties in plume characterization and background $CH_4$. The overall uncertainty for computed $CH_4$ emission rates for an individual determination was estimated to be less than 30% (see Supplemental for a complete evaluation and discussion of uncertainties).

## 3. Results and Discussion

### 3.1 Identification of Sources of $CH_4$

Two example flights from three different facilities (SML, SUN and CNRL) are presented to demonstrate $CH_4$ emissions in the AOSR are mainly from three source types: open pit mining, tailings ponds, and facility activities. Emissions from the remaining two facilities (SAJ and SAU) were shown

to be primarily open pit mining. Source categories were identified by measurements of $CH_4$, $NO_y$, BTEX, and rBC. Figure 2 shows measurements from one low-level transect of a screen flight on Aug 16, 2013 (9 transects in total). This transect was flown at a height of approximately 150 m agl downwind of the SML and SUN facilities, showing clear separation of emission sources from the two facilities. Four distinct plumes are visible, labelled A-D, with linear air parcel back-trajectories

indicated in red arrows. Back trajectories were determined using the wind speeds and wind directions measured on the aircraft at flight level from the positions of maximum $CH_4$, back extrapolated as a general indicator of plume origin. This methodology creates a western bias in our plume origins.  A more careful analysis of surface winds at several meteorological stations in the local vicinity at the time of the aircraft transect shows that surface wind directions were from ~ 140-180o (SE) compared to the

flight level winds, ~220o (SW).  The low level surface winds are likely channelled by the river valley, which runs in a SE to NW direction.  Thus the trajectories of air masses originating at the surface and



mixing upwards have a clockwise rotation, a very local effect, placing the actual plume sources further east than the linear tracks show in Figure 2. Plume A shows a maximum mixing ratio of 2.68 ppm $CH_4$, an enhancement of ~ 0.58 ppm above a background of ~2.1 ppm on this day in this region. This enhanced $CH_4$ is associated with values of 2.3 $\mu g/m^3$ rBC and 47 ppb $NO_y$. The simple linear wind

back-trajectory places the origin of the air mass near the western edge of open pit surface mining activity ~100-min earlier, although as mentioned the actual source is likely slightly east of that location due to the clockwise rotation of the plumes. The combination of rBC and $NO_y$ is indicative of exhaust of heavy hauler diesel trucks that operate in open pit mines. However, significant $CH_4$ emissions are not expected from the truck exhaust, as emissions factors of $CH_4$ from off-road gasoline and diesel

combustion indicate that the $CH_4/CO_2$ emission ratio would be 1 to 2 orders of magnitude lower (Environment Canada, 2015) than the $\Delta CH_4/\Delta CO_2$ observed in this plume (0.58 ppm $CH_4$ / 16.1 ppm $CO_2$). Disturbance of the oil sands at the mine faces by the trucks is a well-known source of $CH_4$ with minor emissions of $CO_2$ and other VOC's (Strausz, 2003) as well as intermediate volatility organic compounds (Tokarek, 2017). Thus, Plume A is interpreted to be a combination of heavy truck exhaust,

indicated by the presence of rBC and $NO_y$, that spatially overlaps with the mine face source of $CH_4$. Plume D shows a similar chemical profile with a maximum $CH_4$ of 2.40 ppm, ~ 0.30 ppm above background, associated with elevated levels of $NO_y$ (40ppb) and rBC (1.5 $\mu g/m^3$). The back-trajectory for Plume D is in agreement with an origin at one of two locations of open pit mining activity at SUN. The two plumes show a similar $\Delta NO_y/\Delta rBC$ ratio within the range of 15-30 ppb per $ug/m^3$. We

consistently measure this profile of $NO_y$ and rBC enhancements from active mines across all five facilities.

Plume B (Fig. 2) shows the highest $CH_4$ mixing ratio at 4.19 ppm, an enhancement of ~2.09 ppm above background. The back-trajectory from the position of the maximum $CH_4$ places the air mass over the

western edge of Mildred Lake Settling Basin (MLSB) tailings pond ~ 20 minutes earlier.  The $CH_4$ enhancement occurs simultaneously with a decrease in $NO_y$ and rBC and an enhancement of total BTEX from ~0 ppb to a maximum of 7.6 ppb. Tailings ponds are known to contain significant quantities of BTEX compounds due to waste streams of mature fine tailings containing naphtha diluent



flowing into the pond (Small et al., 2015). This is similar to the chemical profile observed in Plume C, with a back-trajectory placing the air mass over one of several possible SUN facility tailings ponds shown in Figure 2 (Ponds 6, Pond 5 and Pond 2/3 in figure Small et al, 2015). This indicates the presence of $CH_4$ emissions from multiple tailings ponds. Anaerobic digestion of organic matter in the

tailings pond is the primary mechanism for the production of this biogenic $CH_4$ (Siddique et al., 2012). For Plume C the measured mixing ratio enhancements are 0.25 ppm $CH_4$ and 2.3 ppb of BTEX. The lower $CH_4$ enhancement compared to Plume B suggests less $CH_4$ is emitted from this pond, in agreement with Small et al (2015). The peak-to-peak $CH_4$/BTEX ratios from Plume B and Plume C are ~300 ppb ppb$^{-1}$, and ~100 ppb ppb$^{-1}$ respectively. The difference in measured inter-facility $CH_4$/BTEX

ratios could arise from a number of factors including different pond ages, history, depth, methanogenic behaviour, or use of different diluents in each facility. Our observations are qualitatively consistent with pond-specific industry reported $CH_4$ emission factors, which present SML and SUN Ponds 2/3 (Small et al., 2015) as the highest $CH_4$ emitting tailings ponds in the region. We consistently measured relative enhancements from plumes downwind of SML and SUN according to the pattern of Plumes B

and C in Figure 2, demonstrating the feasibility of using BTEX compounds as tracers for $CH_4$ being emitted from tailings ponds. We expect that BTEX would be greatly reduced from the tailings ponds of those facilities using paraffinic froth treatment (e.g., SAJ) instead of naphtha. In such cases, light hydrocarbons could in principle be used as tracers for the tailings ponds emissions of $CH_4$. However, we did not detect methane plumes above detection limit that were distinct from the open pit mining

plumes of $CH_4$ associated with rBC and NOx for any facilities other than SML and SUN.

Elevated plumes from facility stacks are the primary sources of $SO_2$ in the AOSR due to the bitumen upgrading process. Hence, a significant enhancement of $SO_2$ can be used as a tracer for plant or stack $CH_4$ sources. However, this is not measured at the height shown in Figure 2, which shows a maximum

$SO_2$ of only 5 ppb for this transect between Plumes C and D at 150 m agl. For the same flight (Figure 2), maximum $SO_2$ was 131 ppb for a transect ~350 m above ground, with an associated narrow peak of $CH_4$ with maximum mixing ratio of 2.11 ppm. While higher-altitude $SO_2$ plumes were frequently measured downwind of various facilities over the course of the aircraft campaign, in most cases no





significant CH$_4$ enhancements were observed in these plumes. A similar case is discussed in Section **3.2** where we show the full range of vertical measurements and a lack of enhanced CH$_4$ in the SO$_2$ plume. Ground-level CH$_4$ from tailings ponds and open pit mine faces therefore dominate the CH$_4$ emissions in the region, with minor contributions from industrial plants.

We next compare the profiles from SML and SUN to a third facility, CNRL Horizon. Figure 3 shows a similar transect at ~150 m agl from the Sep 02 flight in the vicinity of CNRL. The bottom panel of Figure 3 shows that there was considerable wind divergence at this time (see back trajectory arrows for A, B, C). This wind divergence was also present in the next pass of the aircraft on the south side of
CNRL at a height of 300m (not shown). While this divergence aids in the visualization of source separation, they invoke uncertainty in the mass balance determinations. The emission rates on Sept 02 were determined using ten transects from a flight much earlier in time than that shown in Fig 3., when the winds were more consistent in direction (NNW).

While Plume A shows a small enhancement of ~1 ppb BTEX downwind of the tailings pond, in contrast to SML and SUN no significant CH$_4$ was associated with it. This is consistent with the pond-specific emission factors presented in Small et al. (2015) that do not list the CNRL tailings pond as a significant source of CH$_4$. The primary Plume B included a CH$_4$ mixing ratio up to 2.24 ppm (enhancement of ~0.34 ppm above background) associated with 12 ppb NO$_y$ and 0.7 ug/m$^3$ rBC downwind of the CNRL
mine. Consistent with the previously described open pit profile and the back trajectory, we identify Plume B as an open pit mining source of CH$_4$. A secondary Plume C was measured with maximum CH$_4$ of 2.02 ppm (enhancement of ~0.12 ppm) east of the open pit mine. The lack of associated species does not relate the origin of Plume C to either a tailings pond nor an open pit mine source of CH$_4$. The plume is downwind of the main CNRL plant and closer in horizontal proximity to SO$_2$ plumes measured
during higher altitude transects. This suggests a CH$_4$ source near the main plant that could originate from venting or flaring activity, electricity cogeneration using natural gas or natural gas leakage. Thus the primary source of CH$_4$ from the CNRL facility is open pit surface mining activity with a secondary undetermined source from the main plant.





Source profiles of $CH_4$ are further compared to measurements of ethane ($C_2H_6$). Source-attribution studies for $CH_4$ commonly use higher ethane-to-methane ratios (EMRs) as a signature for oil and gas emissions, on both a regional (Peischl et al., 2016) and global (Hausmann et al., 2016) scale. $C_2H_6$,

along with other VOC's, was measured from 20-second grab samples collected in 3-L Summa canisters. The VOC's were analysed offline using GC-MS and GC-FID methods described elsewhere [Li et al., 2017]. Table 1 shows $C_2H_6$ measurements from three different flights (Aug 14, Aug 16 and Sep 02) when canister sampling overlapped with the plume descriptions listed previously. In all cases shown, enhancements of $C_2H_6$ above background (0.8 – 1.5ppb) were in the range of only 1-2 ppb, normally the

highest enhancements for each flight (within 1 ppb of 95[th] percentile). The small emissions rates of ethane (EMRs <1.4% ) across flights contrasts with the high EMRs (i.e. 40-50%) seen for conventional oil and gas fields in other regions of North America (Peischl et al., 2016) and is lower than all the possible EMR source scenarios tested in Hausmann et al. (2016). The low EMRs are consistent with previous measurements in the region (Simpson et al., 2010) and are an indication of the unique

character of unconventional bitumen sources. As such, global estimates of the relative contributions of oil and gas emissions to decadal increases in atmospheric $CH_4$ that are based on $C_2H_6$ and $CH_4$ measurements in the free troposphere (Hausmann et al., 2016) would not capture AOSR emissions due to the low $C_2H_6$ emissions in this region.

## 3.2 Quantification of CH₄ Emission Rates from Sources

The source chemical profiles in section 3.1 can be used in combination with the screen mass-balance method (Eq. 1) to isolate and quantify categories of AOSR emissions. As an example, we show the Aug 14 flight surrounding the SML facility, which consisted of a box and screen path flown in rapid succession. Figure 4 shows an image of the interpolated aircraft measurements from the box path creating a contiguous mesh superimposed on a map of the region. Winds were from the south at 186 ±

48 degrees over the course of the day. Three distinct ground-based plumes of $CH_4$ are visible, a primary plume (Plume N) on the northern screen (~6500 m wide) exiting the box, a secondary plume (Plume NW) at the northwest corner (~7000 m wide) exiting the box and a smaller plume (Plume E) on the





eastern screen (~3000 m in width) entering the box from outside the SML facility boundary. The lowest aircraft transect was at a height of ~150 m agl, with maximum $CH_4$ mixing ratios of 3.00 ppm, 2.60 ppm and 2.63 ppm respectively for the three plumes. Mixing ratios of $CH_4$ below 150 m agl, are based on a linear extrapolation of interpolated pixels to the surface, corresponding to maximum surface

mixing ratios of 3.48, 3.17 and 3.06 ppm for the primary (N), secondary (NW) and tertiary (E) plumes respectively. Extrapolation to the surface is the primary source of uncertainty for surface sources in this method, however the uncertainty can vary between flights depending on the meteorological conditions (Gordon et al., 2015). As a part of our uncertainty analysis in the Supplemental material, we have included an uncertainty associated with the differences in emission rates that arise from the use of

linear, constant and half-Gaussian extrapolations in the calculations.

Unwrapped curtain plots of $CH_4$, BTEX, $NO_y$, rBC and $SO_2$ from the Aug 14 box flight (Fig 4) are shown individually in Figure 5, projecting the 3-D virtual box onto a 2-D grid. The same three plumes from Fig 4 are highlighted by dotted boxes in red (N screen), yellow (NW corner), and black (E screen).

The red and yellow boxes show sources originating from within the SML facility and the black boxes show a source originating outside of the SML boundaries and entering the box from the east. The largest SML $CH_4$ plume is associated with > 10 ppb BTEX and the absence of rBC and $SO_2$, with some $NO_y$ (~20 ppb). This is consistent with the chemical signature associated with tailings pond emissions discussed previously. The NW plume is associated with >60 ppb of $NO_y$ and up to 5 $\mu g/m^3$ of rBC, with

minimal BTEX and $SO_2$, consistent with the expected chemical signature from open pit surface mining. The smaller plume on the E screen is associated with elevated BTEX and $CH_4$ and is likely a plume from one of the SUN tailings ponds as the winds indicate the plume is entering the box. The elevated plume in Fig 5 (orange circles) with ~100 ppb $SO_2$ and ~30 ppb of $NO_y$ is traced to the SML upgrader activities, but with no enhancement of $CH_4$ above background on this day. A second $NO_y$ plume is

visible at the north-eastern corner of the box, not associated with any of the identified $CH_4$ source types. This $NO_y$ plume likely originates from traffic on the main highway that passes between the SML and SUN facilities and/or trucks and other vehicles operating in and around the main SML facility.





Boundaries of the plumes from separate sources are estimated using the tracer species listed in Fig. 5 by evaluating where the chemical signatures reached background levels. However, the SML tailings pond and open pit mine plumes were not completely resolved from one another, overlapping within a range of ~800 m.  The uncertainties in the emission rates due to plume overlap were estimated by contracting and expanding the horizontal integration boundaries (s) by 800 m on each side (a total of ±1600 m) as part of the sensitivity analysis. A vertically varying background profile ($[CH_4]_B(z)$) is determined using data from the upwind southern screen, as mentioned previously. Using a spatially identical screen of perpendicular wind $U_⊥(z)$, the fluxes are determined through each pixel and the total source emission is calculated by integrating the pixels within the plume boundaries (Eq. 1). $CH_4$ emissions rates are computed to be $6.4 \pm 1.2$ metric tonnes per hour (tonnes hr$^{-1}$) for the SML main tailings pond and $2.7 \pm 0.6$ tonnes hr$^{-1}$ for the SML open pit mine source.  It is possible that the $CH_4$ plume from the SML tailings pond includes $CH_4$ emissions from the main SML plant facility (flaring, venting, natural gas leakage, etc.) that cannot be spatially separated from one another due to their proximity, however we anticipate the magnitude of these emissions are minor and captured within the error intervals listed.

This screen-based mass-balance approach for determining specific source emission rates (Eq. 1) is applied to flights with appropriate conditions for plume separation. Mean emissions rates of $CH_4$ from specific sources within the facilities SML, SUN, CNRL, SAU and SAJ are shown in Figure 6. SML emissions rates are the average of three mass-balance flights over two days (two on Aug 14 and one on Aug 16). Two flights on separate days were used for each of the SUN (Aug 16 and Aug 29), SAJ (Aug 21 and Sep 06) and SAU (Aug 29, Sep 06) facilities. One CNRL flight (Sep 02) had northerly wind conditions showing plume separation on a southern screen. No significant daily variability is observed as the emissions rates for the same source agree within error. Duplicate and triplicate estimates for the same source are combined using an error-weighted uncertainty (Supplemental). SML and SUN had significant open-pit mining emissions of $CH_4$, $2.8 \pm 0.4$ tonnes hr$^{-1}$ and $1.8 \pm 0.2$ tonnes hr$^{-1}$ respectively, and were the only facilities with tailings ponds emissions above detection limit, $6.4 \pm 0.8$ tonnes hr$^{-1}$ and $2.4 \pm 0.3$ tonnes hr$^{-1}$. CNRL had open-pit mining emissions ($2.6 \pm 0.7$ t/hr) and



significant emissions originating from the main plant facility ($1.0 \pm 0.3$ t/hr). Plumes of $CH_4$ from SAJ and SAU were only attributed to open-pit emissions, $1.2 \pm 0.2$ t/hr and $1.4 \pm 0.2$ t/hr respectively.

The plume-targeting screen mass balance method described here is unable to resolve emissions of $CH_4$ from multiple sources not characterized by the chemical profiles described in Section 3.1 if they cannot also be spatially separated. Because spatial s and z constraints are manually chosen by plume boundaries from chemical profiles, minor emissions may contribute to overestimation of the emissions from an individual source when highly coincidental in space such that the sources are not separable. For example, the emissions from the main plant were identifiable in the case of CNRL due to the separation and orientation of the plant, the open pit and the tailings ponds with respect to the winds. This was not the case for the other major facilities in the AOSR where many of the sources were highly coincidental in space. It is possible, and even likely, that other major facilities in this study also have $CH_4$ emissions from their main plants (venting, cogeneration, natural gas leakage, etc.) that are identified as tailings pond emissions or open pit emissions due to close proximity and our inability to deconvolute the sources spatially or chemically. However, we expect that the total emission rates of $CH_4$ from each facility are still accurate.

Emissions rates from each flight and individual sources (where possible) using the screen mass balance method are tabulated in the supplemental (Tables S1-S5). We did not measure a detectable tailings pond source of $CH_4$ from CNRL, SAJ and SAU. Associated enhancements of rBC and $NO_y$ with $CH_4$ suggest that the $CH_4$ source from SAJ and SAU is also predominantly open pit mining. The results using the screen mass balance approach (Eq. 1) are further verified in Section 3.3 using emissions rates for each facility determined from the box approach (Eq. 2).

### 3.2.1 Comparison to Fugitive Emissions Literature

Average open pit surface mining emissions from the five facilities are within a range of 1.2-2.8 tonnes hr$^{-1}$ (Fig 6 and Table S1-S5). This shows some consistency in the nature of $CH_4$ release from open pit mining activity in the region, with differences that may possibly be attributed to the size of the surface





disturbance taking place and the intensity of the mining activity. Methane emissions from open pit mines were recently estimated using a bottom-up emissions factor approach by analysing the gaseous composition in the overburden and oil sand component of drill core samples (Johnson et al., 2016). Emissions factors of $CH_4$ were then scaled up according to the total mass of material mined or the total

bitumen produced. For 2013, Johnson et al. estimate total fugitive mining emissions in the region to be 21.4-46.0 ktonnes of $CH_4$ using total mined material, and 33.1-85.0 ktonnes of $CH_4$ using total mined bitumen. Our top-down approach estimates total fugitive emissions from open pit mining to be $9.7 \pm 0.9$ tonnes hr$^{-1}$, corresponding to $84.9 \pm 7.9$ ktonnes yr$^{-1}$ $CH_4$ if constant temporal emissions are assumed. Agreement with the upper estimates in Johnson (2016), despite the uncertainty associated with

extrapolation to annual emissions, suggests that their bottom-up emissions factors from gases in core samples may reliably predict real-world emissions provided there is accurate characterization of $CH_4$ in the core samples over the entire disturbed area. This is reasonable considering oil sands degassing tends to release $CH_4$ quantitatively in a predictable way.

From our 2013 measurements, only two facilities, SML and SUN, had significant emissions of $CH_4$ from tailings ponds. Tailings ponds emissions accounted for ~70% and ~58% of total $CH_4$ from SML and SUN respectively. This accounted for ~45% of total emissions in the AOSR. Recently, bottom-up area-weighted emissions factors of $CH_4$ from 19 major tailings ponds in the AOSR were provided for the year 2012 (Small et al., 2015). The top three emitting ponds reported were 'Mildred Lake Settling

Basin' (MLSB) and the 'West In-Pit' (WIP) pond within SML, and 'Pond 2/3' (P23) within SUN. These tailings ponds account for >96% of tailings ponds $CH_4$ in the region according to that study. This is qualitatively consistent with our measurements of $CH_4$ mainly from SML and SUN. Our method requires $CH_4$ plumes to be clearly enhanced above background, so trace amounts of $CH_4$ from less active younger ponds in the other three facilities were not detected. We are unable to differentiate

emissions from ponds within the same facility due to overlapping chemical profiles from ponds within close proximity. However, using the ratios of relative pond emissions rates within the same facility presented in Small et al. (2015), (i.e. MLSB contributes 92% to SML emissions, Ponds 2/3 contribute 85% to SUN), we can infer individual pond emissions from our measurements assuming the relative



contributions are accurate. The resulting emissions rates are $5.8 \pm 0.8$ tonnes hr$^{-1}$ for MLSB and $2.0 \pm$ 0.3 for Ponds 2/3. This ranks the MLSB tailings pond as the highest area source of $CH_4$ in the AOSR, followed by the open-pit mines in SML and CNRL, and fourth Ponds 2/3 in SUN. Total $CH_4$ from tailings ponds in Small et al. (2015) are estimated to be 30.3 ktonnes of $CH_4$ per year, with 29.7 ktonnes of $CH_4$ from the SML and SUN facilities (~98%). Our total $CH_4$ emission rate determined for tailings ponds is $8.8 \pm 0.9$ tonnes hr$^1$, which corresponds to $77.1 \pm 7.9$ ktonnes yr$^{-1}$ if a constant temporal factor is assumed. This is 2.3-2.9 times larger than the emissions inferred from the data in Small et al (2015), despite the uncertainty of extrapolation to an annual emissions rate. Our measurements suggest more work is needed to reconcile top-down and bottom-up $CH_4$ emissions.

## 3.3 Emission Rates of $CH_4$ from AOSR Facilities

Total emissions rates of $CH_4$ from each facility determined using the box mass balance method (TERRA) are tabulated in the supplemental (Tables S1-S5) along with the determinations using the screen approach. Where multiple screen estimates or multiple box estimates were available, uncertainty weighted ($1/\sigma^2$) averages were determined for each method for each facility and are summarized in Table 2. While the box method is in some cases based on the same downwind measurements as the screen approach, the two methods have several key differences (described in **2.3**) and are treated as independent estimates. In particular, the box method does not resolve specifically targeted, individual plumes and instead determines the net outgoing flux from the closed volume surrounding the facility. Thus, consistency between the two estimates is evidence that the primary sources of $CH_4$ from facilities in the AOSR are tailings ponds, open pit mines and facility emissions captured by the source characterization in sections 3.1 and 3.2. In general, the total emissions from each facility using the screen and box methods are in agreement within uncertainty, which adds confidence to the measured emission rates reported here. In the final row of Table 2, we calculate a weighted average emission rate for each facility using all screen and box measurements. The $CH_4$ emission rates from the facilities are $8.6 \pm 0.9$, $4.2 \pm 0.4$, $3.6 \pm 0.5$, $1.3 \pm 0.2$, and $1.5 \pm 0.2$ tonnes $CH_4$ hr$^{-1}$ from the SML, SUN, CNRL, SAJ and SAU facilities, respectively.



### 3.4 Total Emissions of CH₄ from the AOSR

The total $CH_4$ emissions from the five mining facilities in the AOSR, obtained by summing the best estimates (i.e., uncertainty weighted average of multiple measurements, bottom row, Table 2) of the individual facility emission rates is given in the final row and column of Table 2. The 5-facility total

emission rate of $CH_4$ is $19.2 \pm 1.1$ tonnes $CH_4$ hr$^{-1}$. A final independent estimate of total AOSR emissions was obtained from a flight on Aug 16, utilizing an independent transect screen ~75 km wide (Aug 16 Screen B) downwind of all major mining facilities in the AOSR (excluding Imperial Kearl Lake and Suncor Firebag operations; but also inclusive of any emissions from Suncor MacKay River in-situ facility). The details of this flight are given in Supplemental Table S6. The interpolated screen from

the Aug 16 flight (Total OS) is shown in Figure 7. The screen was constructed from 10 aircraft horizontal transects from 250-900 m agl. Enhancements of $CH_4$ were measured over a wide horizontal subrange of ~60 km of the entire ~75 km screen. Winds were perpendicular to the plane from the southwest (225°), showing a large flux of $CH_4$ through the screen from upwind sources. The highest measured mixing ratios of $CH_4$ were 2.67 ppm at the ~250 m (agl) transect. Background $CH_4$ in the

region was ~2.00 ppm taken as a vertical profile from the wings of the screen. Using the screen method (Eq. 1), the emissions rate was determined to be $23.0 \pm 3.7$ tonnes $CH_4$ hr$^{-1}$, which represents the emissions from all major facilities within the AOSR domain. This AOSR total is only slightly larger than the previous 5-facility total emission rate of $19.2 \pm 1.1$ tonnes hr$^{-1}$, but not statistically so, demonstrating the reproducibility of our measured estimates. It is entirely possible that there are other

minor sources of $CH_4$ included in this larger number from smaller industrial operators in the region, trucks and vehicles on the main highway, and wetland emissions. In fact, the Canadian GHGRP inventory (see section 3.5) indicates that there is an additional 0.13 tonnes $CH_4$ hr$^{-1}$ emitted upwind and 0.17 tonnes $CH_4$ hr$^{-1}$ emitted downwind of the aircraft screen (Fig 7) from minor industrial facilities within the AOSR (both numbers downscaled from the facility reported annual emissions). The amount

of $CH_4$ emitted from vehicles on the highway though is expected to be smaller. The fact that the two numbers are not statistically different supports the determination that the majority of the $CH_4$ in the AOSR is emitted from the 5 major industrial facilities in the region. The two values are combined here



using an error-weighted uncertainty resulting in a final AOSR facility emissions estimate of $19.6 \pm 1.1$ tonnes $CH_4$ $hr^{-1}$.

## 3.5 Comparison to Emission Inventories

Emissions of anthropogenic greenhouse gases are estimated by ECCC in Canada's GHG Inventory, which forms the basis for Canada's annual report to the United Nations Framework Convention on Climate Change , UNFCCC (ECCC, 2016). Currently, industrial facilities that emit more than 50 ktonnes $CO_2eq$ $yr^{-1}$ are required to report their emissions annually to ECCC using the Greenhouse Gas Reporting Program (GHGRP), which is Canada's legislated, publicly accessible inventory of facility-reported greenhouse gas (GHG) data (ECCC, 2017a). Although the GHGRP inventory data is not necessarily used in Canada's GHG Inventory, changes are being proposed to expand monitoring requirements in the GHGRP, including lowering the reporting threshold to 10 ktonnes $CO_2eq$ $yr^{-1}$ in order to enable direct use of the reported data in Canada's GHG Inventory (ECCC, 2017b). Emissions of $CH_4$ from all five major oil sands facilities discussed in this paper are present in the GHGRP Emissions Inventory on an annual basis. The annual emission rates of $CH_4$ extracted from the inventory were downscaled to hourly emissions rates for comparison with our measurements with an assumption of equal seasonal and diurnal profiles 365 days a year, 24 hours per day; for consistency with upscaling factors used to generate annual emissions. While this may be questioned, it should be noted that fugitive emissions of $CH_4$ from mine faces and tailings ponds in the inventories are estimated based upon emission factors measured at oil sands facilities during summer months (June- Sept), which are then up scaled from hourly emissions to annual emissions using the same assumption that we used to downscale (365 x24), as per recommendation by the Government of Alberta (GOA, 2014). Specifically it is noted from the GOA report that emissions of gaseous species such as $CH_4$ (and $CO_2$) are not temperature dependent, (unlike VOC's that have temperature dependent vapour pressures (Li et al., 2017)). One argument for the use of the constant temporal factor is that temperatures at depth in a tailings pond, rightly or wrongly, are said to remain relatively constant throughout the year (GOA, 2014) and thus biogenic gas formation continues in the winter. For mine faces, the GHG component of the oil sand does not change with temperature and is likely released completely in a short period of time




after being mined. Thus the government recommendation to oil sand facilities in preparing annual emission estimates of fugitive GHG's is that reduction factors should not to be used in extrapolating summertime emissions over the rest of the year (GOA, 2014). Figure 8 shows a comparison of the total measured emission rates of $CH_4$ from the five industrial facilities (2013), the total measured $CH_4$

emission rate in the AOSR from the single downwind screen on Aug 16, 2013 and the sum of the facility emission rates from the Canadian GHGRP Emissions Inventory for 2013, expressed in hourly units. The combined facility emissions rate of $19.6 \pm 1.1$ tonnes $hr^{-1}$ is approximately $48 \pm 8$ % higher than the 5-facility total of 13.2 tonnes $hr^{-1}$ extracted from the inventory for 2013. Facility-to-facility comparisons show higher measured than reported emission rates for three out of the four facilities (SML

and SAU facilities are combined as one in the inventory). In contrast, for CNRL our measured emission rate is 1.2 tonnes $hr^{-1}$ lower than the inventory. Since we have determined the composition of SML, SUN and SAJ emissions to be primarily from tailings ponds and open pit mining, there appears to be underestimation in the inventory of those particular area sources within these sites.

These discrepancies indicate a need for inventory reconciliation between the bottom up and top-down

estimates. It has been shown possible to reconcile divergent bottom-up and top-down $CH_4$ estimates for the Barnett Shale by using more comprehensive activity factors and better characterization of emissions from high emitter sites (Lyon et al., 2015) and continuous monitoring to identify these super emitters (Zavala-Araiza et al., 2015). Currently, bottom-up estimates in the AOSR are accomplished by systematic surface flux chamber measurements of area sources (surface mines, tailings ponds) to derive

area-based emissions factors (GOA, 2014). While surface flux chamber measurements (Klenbusch, 1986;Conen and Smith, 1998) are estimated to be 50-124% of the true emissions rate for a homogenous source (Klenbusch, 1986), it is unclear how the uncertainty propagates when the emissions factors are scaled to the full surface area of the heterogeneous AOSR emissions sources. The official survey protocol for open pit sources attempts to minimize the possibility of underestimating emissions by

explicitly requiring fugitive surveys to include sampling at a range of locations within the open pit mine, where safe to do so, including high priority zones (disturbed in the last week), normal priority zones (disturbed from 1 week to 6 months ago) and low priority zones (disturbed > 6 months ago) (GOA, 2014). However, it seems that the recent methodology outlined by Johnson et al. has great



promise and reduced uncertainty (Johnson et al., 2016) for estimating fugitive emissions from open pit mining.

## Conclusions

We present a detailed approach to identifying and quantifying $CH_4$ emission sources from the surface
mining facilities in the Athabasca Oil Sands Region of Alberta in the year 2013. Emissions of $CH_4$ are
attributed to three major fugitive source types: tailings ponds, open pit mining activity, and emissions
from plant facilities. Our method demonstrates the use of BTEX/VOCs as tracers for tailings ponds $CH_4$
plumes due to the use of diluent, and $NO_y$/rBC as tracers for surface mining due to heavy hauler diesel
trucks operating co-spatially at mine faces in the open pit mines. The combination of $SO_2$/$NO_y$ is used
as a tracer for stack facility plumes which are observed to contain minor but detectable quantities of
$CH_4$, although infrequently. We use the chemical signatures of sources and the screen mass-balance
approach for 7 flights to determine total emissions rates of $8.8 \pm 0.9$ tonnes $hr^{-1}$ from tailings ponds,
45% of total $CH_4$ emissions in AOSR, $9.8 \pm 0.9$ tonnes $hr^{-1}$ from open pit surface mining (50%), and 1.0
$\pm 0.3$ tonnes $hr^{-1}$ primary facility-associated and other sources (5%). Open pit mining emissions are
measured from all five facilities in the range of 1.2-2.8 tonnes $hr^{-1}$. In contrast amongst the 19 tailings
ponds in the region, $CH_4$ emissions above determinable levels were only measured from two facilities,
SML and SUN. These emissions are likely due to two tailings ponds, MLSB ($5.8 \pm 0.8$ tonnes $hr^{-1}$) and
Ponds 2/3 ($2.0 \pm 0.3$), which are ranked amongst the highest area sources of $CH_4$ in the region. These
results demonstrate the large contributions (~45%) of a few tailings ponds sources to total fugitive $CH_4$
emissions in the AOSR and highlight opportunities for strategic GHG mitigation. Our individual-plume
sum is consistent with estimates derived using the TERRA box approach to determine total emissions
within facility boundaries. The agreement between these two methods demonstrates that the three
source types listed are representative of the major emissions of $CH_4$ in the AOSR. Further results from a
~75km flight screen that captured amost all AOSR emissions are able to reproduce total emissions
derived from the sum of the five major facilities. Our final top-down estimate of the 2013 summer time
emission rate in the region is $19.6 \pm 1.1$ tonnes $CH_4$ $hr^{-1}$ or $0.17 \pm 0.1$ Tg $CH_4$ $yr^{-1}$. We note that the
annual emissions rate is only a first order approximation of what annual emissions might be if the





temporal emissions are constant throughout the year; however, we consider this assumption to be highly uncertain as the seasonality of fugitive emissions rates of $CH_4$ is still a major uncertainty. Further effort should be devoted to measurements of these emission rates in different seasons, and to understand if ambient temperature and ice coverage on tailings ponds are important parameters or not. Our limited

measurements of ethane and methane downwind of the AOSR facilities suggest that the EMR's are quite low, $< 1.4\%$, likely because the fundamental source of the majority of the methane emissions are methanogenic, not thermogenic, in nature. Thus global estimates of the relative contributions of oil and gas emissions to increases in atmospheric $CH_4$ based on EMR measurements in the free troposphere would not capture AOSR emissions due to the low $C_2H_6$ emissions in the region.

**Acknowledgements**

Funding for the measurement campaign and the subsequent analysis was provided by the Climate Change and Air Pollution program of Environment and Climate Change Canada and from the Canada-Alberta Joint Oil Sands Monitoring Program. We thank the Convair-580 flight crew of the National Research Council of Canada, especially the pilots (Paul Kissmann, Rob Erdos and Tim Leslie), for

conducting the aircraft flights. We thank the technical support staff, especially Andrew Sheppard, and the Data Management Team of the Air Quality Research Division for their hard work and support throughout the aircraft campaign. We thank Stewart Cober for his review of this manuscript, management in the field and skills as a flight director. RM and SB acknowledge funding from NSERC Discovery and a grant from ECCC to support some of this work. SB & RM also acknowledge funding

from the NSERC CREATE program, Integrating Atmospheric Chemistry and Physics from Earth to Space (IACPES).

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



**Table 1:** Enhancements of $\Delta C_2H_6$ from canister measurements overlapping with $CH_4$ plumes across three flights (Aug 14, Aug 16 and Sep 04). Mean enhanced $\Delta CH_4$ is shown over the course of ~20s canister sampling times with ethane-to-methane ratios (EMRs) computed.

| Scenario | $\Delta C_2H_6$ (ppb) | $\Delta CH_4$ (ppb) | EMR (%) |
|---|---|---|---|
| SML Ponds (Aug 14) | 3.2 | 814 | 0.40 |
| SML Mines (Aug 14) | 2.6 | 365 | 0.72 |
| SUN Ponds (Aug 16) | 1.2 | 215 | 0.56 |
| SUN Mines (Aug 16) | 1.1 | 185 | 0.59 |
| CNRL (Sep 04) | 1.9 | 137 | 1.39 |

**Table 2:** Comparison of emissions rates (in tonnes $CH_4$ $hr^{-1}$) determined from the screen approach (n estimates per facility), the box-approach (n estimates per facility), and the uncertainty weighted average for each method and facility. The 5-facility AOSR total is show in the final column and row.

| Method | SML (n) | SUN (n) | CNRL (n) | SAJ (n) | SAU (n) | Total AOSR |
|---|---|---|---|---|---|---|
| Screen | 9.1±0.9 (3) | 4.2±0.4 (2) | 3.6±0.8 (2) | 1.2±0.2 (2) | 1.4±0.2 (2) | |
| Box | 7.7±1.5 (1) | 3.9±0.9 (1) | 3.6±0.6 (2) | 1.4±0.2 (2) | 1.7±0.3 (1) | |
| Average | 8.6±0.9 (4) | 4.2±0.4 (3) | 3.6±0.5 (4) | 1.3±0.2 (4) | 1.5±0.2 (3) | **19.2 ± 1.1** |





**Figure 1:** Flight tracks from flights capturing emissions from SML (Aug 14, Aug 16), SUN (Aug 16, Aug 29), CNRL (Aug 20, Sep 02), SAJ (Aug 21, Sep 06 not shown), and SAU (Aug 29, Sep 06 not shown). SML and SAU are shown in blue, SUN in pink, CNRL in yellow and SAJ in dark orange.





**Figure 2:** Top: Aircraft measurements of $CH_4$ (red), BTEX (blue) and rBC (black) from a single transect at 150 m agl downwind of SML and SUN on Aug 16. Four plumes are labelled A (SML Mine), B (SML Tailings), C (SUN Tailings), D (SUN Mine). Bottom: $CH_4$ mixing ratios along the 150 m agl transect for the above time series. Red arrows indicate air parcel back-trajectories based on linear back extrapolation of flight level measured wind vectors at plume centres, with end points at 100 min (A), and 20 min (B-D).





**Figure 3:** Top: Aircraft measurements of $CH_4$ (red), BTEX (blue) and rBC (black) from single transect
~150 m agl downwind of CNRL. Plume A (CNRL Tailings Pond), Plume B (CNRL Mine) Plume C
CNRL Facility. Bottom: $CH_4$ mixing ratios along 150 m agl transect for above time series. Red arrows
show back-trajectories based on linear extrapolation of measured wind speed and direction.



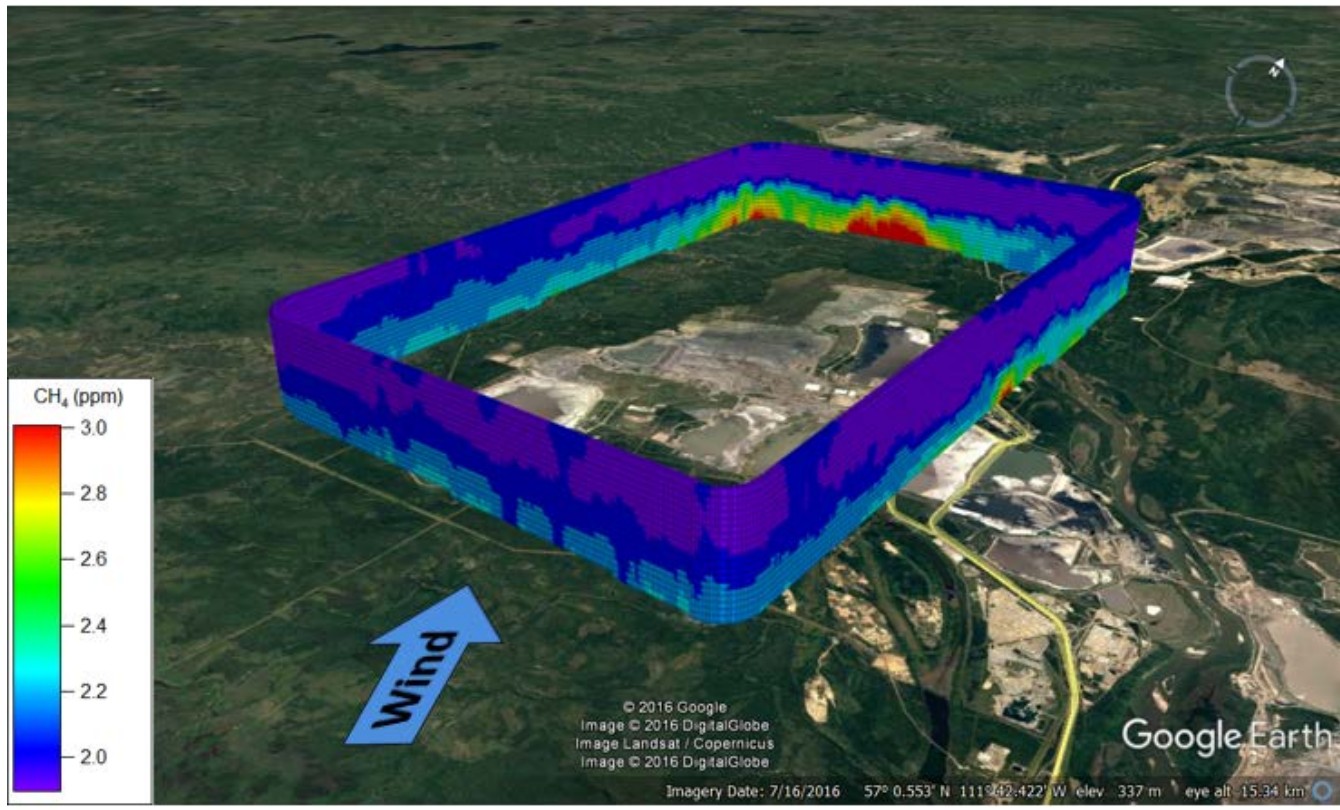

**Figure 4:** Interpolated CH$_4$ mixing ratios for the Aug 14 box flight surrounding SML.



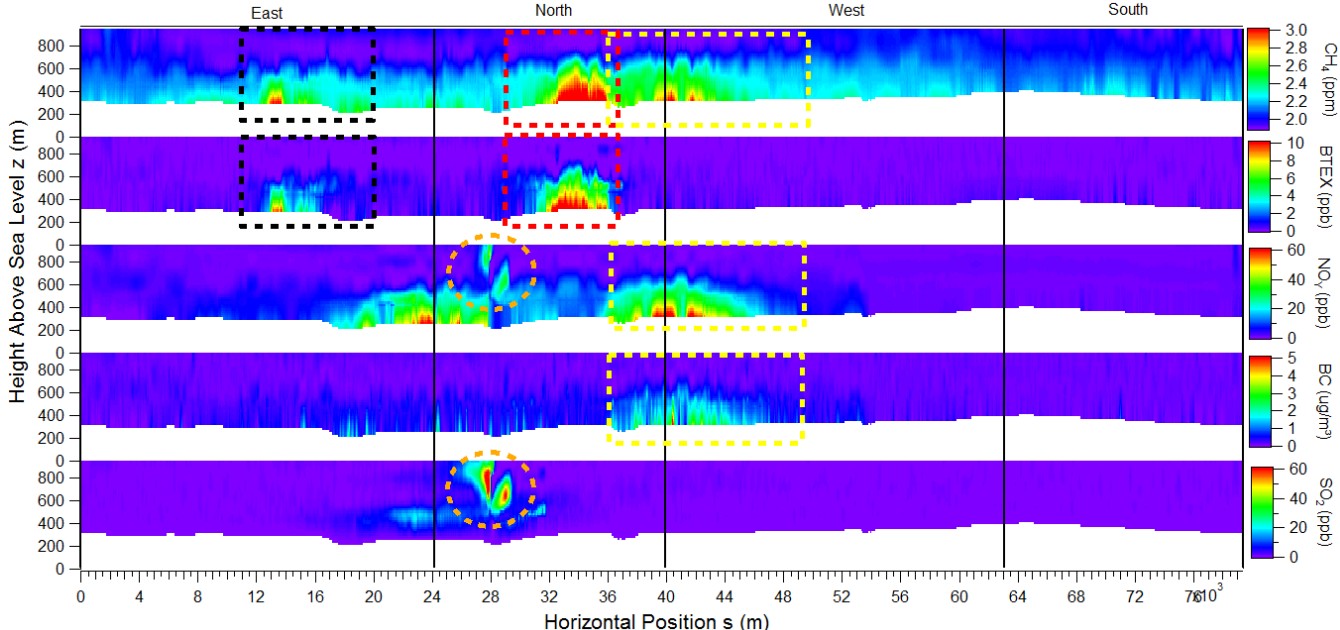

**Figure 5:** Curtain plots showing interpolated CH$_4$, BTEX, NO$_y$, rBC and SO$_2$ mixing ratios for the Aug 14 box flight around SML. Red-dashed box indicates the primary plume on the North screen, yellow-dashed box indicates the secondary plume on the West screen, and black-dashed box indicates the tertiary incoming plume on the East screen. Orange dashed-circle shows the upgrader plume on the North screen.



**Figure 6:** Source-apportioned emissions rates of CH₄ determined by the screen mass-balance method
(Eq. 1) for the SML, SUN, CNRL, SAJ and SAU facilities. Emissions rates are the average of three
mass-balance flights for SML over two days, two flights each for SUN, SAJ and SAU on separate days,
and one flight for CNRL.



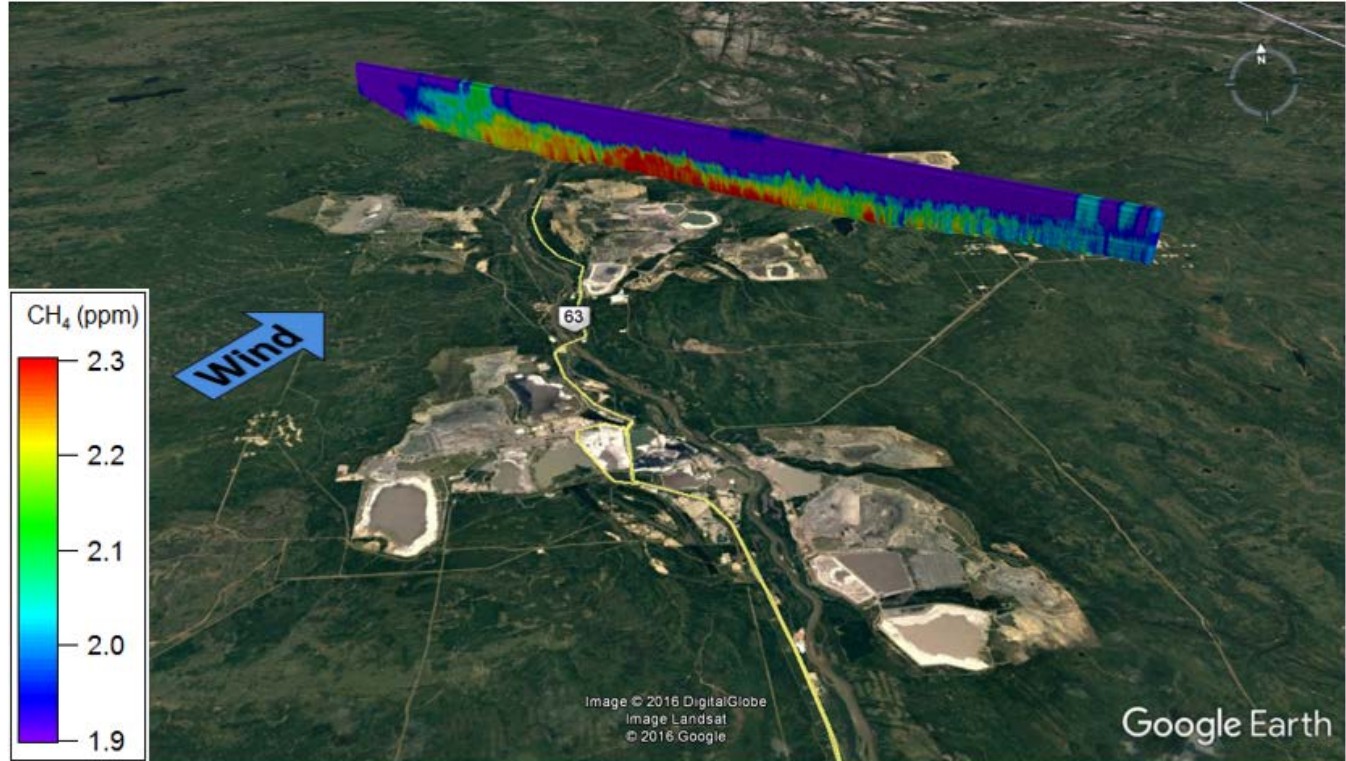

**Figure 7:** Map image showing interpolated CH$_4$ mixing ratios for the Aug 16 Total Oil Sands screen.





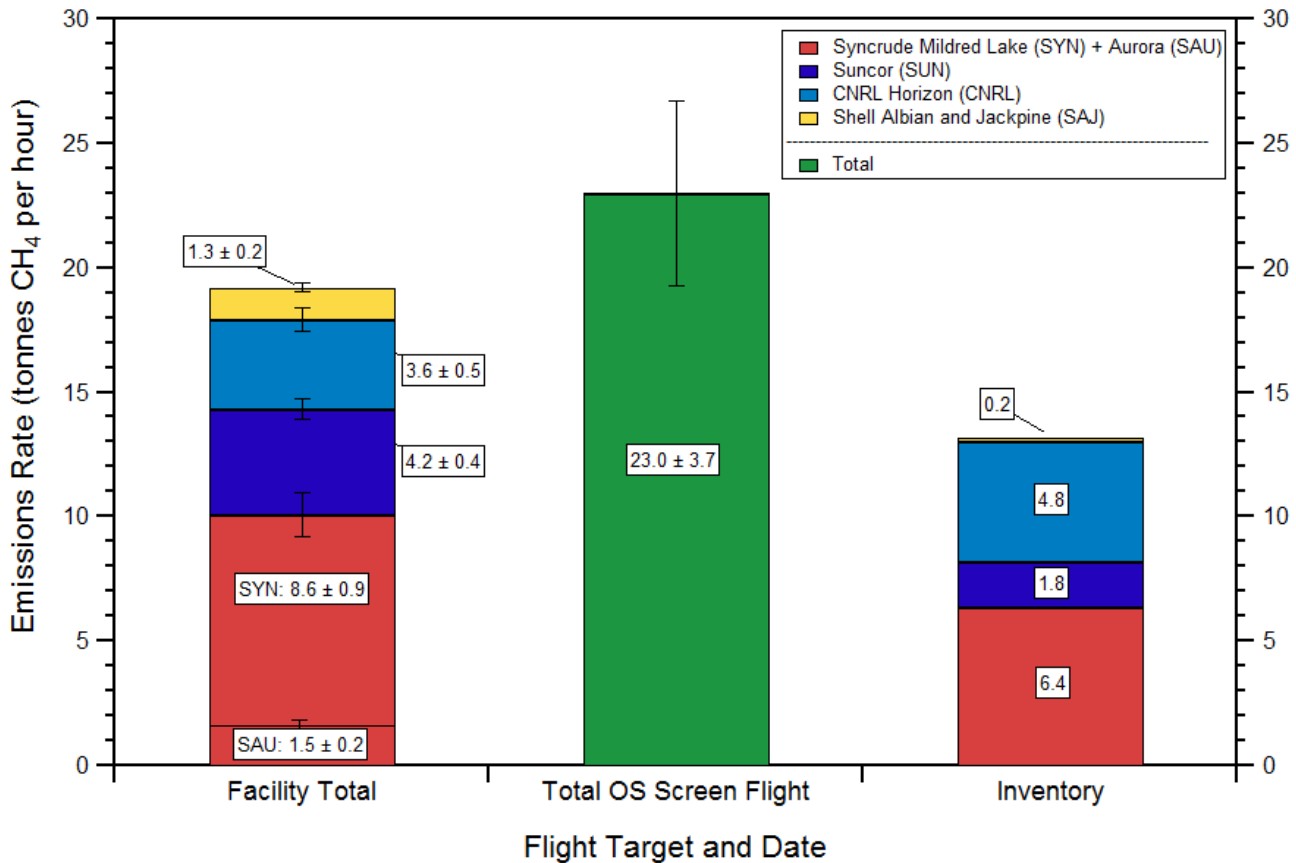

**Figure 8:** Comparison of emissions rates determined for the five major facilities (SML, SAU, SUN, CNRL, SAJ, SAU) with the Total OS Screen Flight (see Fig. 7 for the flight track). Also shown is the CH$_4$ emissions taken from the Canadian GHGRP Emissions Inventory for the year 2013, scaled down from annual to hourly emissions assuming constant temporal emissions. Note that in the inventory, SML and SAU emissions are reported as a single facility, while our estimates are derived separately.