# Peer review of "Ouantification of Methane Sources in the Athabasca Oil Sands Region of Alberta by Aircraft Mass-Balance"

_Atmospheric Chemistry and Physics, 2017_

## Referee Comment (RC1) · Anonymous Referee #1 · 5 Jan 2018

Summary/General comments: Baray et al. present aircraft measurements made around the Athabasca Oil Sands region and employ multiple mass balance approaches to quantify methane emissions from the entire region as well as individual facilities/components to the region. They also use multiple trace gases to attribute to specific processes, and compare results with reported inventoried emissions. This paper is well placed in ACP. This paper contributes to our understanding of methane emissions from a unique but potentially high impact source region. Overall this is a well-written paper, a very nice/ sound analysis, and I enthusiastically recommend publishing with only a few minor suggestions.

[Figure]

Minor comments: Page 4, lines 1-20: this introduction portion is long and dedicated to the recent confusion about global methane and global methane trends. While accurately written, I don't think it is helpful for this paper. Addressing methane emissions from the oil sands is not going to help with these large questions, and motivating the oil sands emissions does not need invoking some the global decadal confusion, but instead could be better motivated focusing on the work in the last 10 years attempting to address methane emissions from the oil and gas production sector, where large discrepancies have been found and this work contributed nicely.

Throughout: Please change the units for methane from ppm to ppb. It is standard to work with methane in ppb, and as the signals observed and discussed make more sense to see in ppb than ppm, this change should be made throughout.

Figure 1: Would help a lot to have spatial scale on these figures. Also would be useful to have some wind arrows indicating what winds look like on each of these flight days.

Figure 3 (and applies to other plumes): I would like to see what the correlation looks like between different gases within each designated plume. Some tracer-tracer plots with the different plumes shown would be helpful to show/establish how robust the correlations are for each of these tracer-tracer relations.

Page 16, Lines 1-18 as well as Table 1: I'm a little worried about the ethane:methane analysis and would like more supporting information. Smith, Kort, Karion et al., 2015 ES&T used continuous ethane:methane measurements over the Barnett Shale and showed that using limited, discrete flask samples could lead to erroneous ethane:methane ratios. It would help if the authors showed on the time series plot illustrating the plume where in the plume(s) the flasks were collected to help illustrate what the flask ethane may be representative of. The limited discrete samples may have been sufficient, or there may be important gaps causing an uncertainty in how much ethane in fact was emitted – at this point I cannot assess this and this should be improved.

Page 18 Line 6: The vertically varying background can be troublesome/worrisome. It would be helpful to see the profile that is used here and understand how variable the background is.

Page 26, lines 1-2: Should specify the seasonality of fugitive emissions from this unique oil sands source are unknown, not fugitive emissions in general.

---

## Referee Comment (RC2) · Anonymous Referee #2 · 12 Jan 2018

This paper presents a thorough study of emissions from oil sands facilities. While methane is the focus of these aircraft measurements, a number of complimentary species help to characterize emissions and separate individual sub-sources at each site.

The authors do a nice job of contrasting the emissions from the different facilities visited, and bring in previous measurement and inventory work for context.

The paper is well-written and organized. The curtain and box methodologies are accurately and simply described. I believe this paper is appropriate for publication in ACP, with only a few minor edits:

The first two and a half pages of introductory material discuss methane and its climate and ozone formation impacts. I think that this background material should be condensed, with more of a focus on the oil sands region.

The introductory material starting on page 5, line 11 is of utmost interest to this study. I recommend this section be supplemented with a sentence or two about anaerobic methane formation in tailings ponds, which is mentioned briefly later (p. 14 line 5).

Related to the above comment, on p. 20 line 24, the authors note that younger ponds should produce less methane. The subject of tailing pond methane emissions warrants a paragraph of discussion in the text, explaining why the high emissions from P23 and WIP might be expected (age, any other process differences), and why emissions of methane were low/undetected from other ponds.

p.14, line 19: "we did not detect methane" : Were any canisters taken showing light hydrocarbon enhancements?

p. 20 line 12: Is there a reference or previous study that looks at this degassing rate?

p. 22 line 19, and elsewhere: When discussing seasonality of emissions, it would be useful to remind the reader that these measurements were taken in August-September

p. 23, line 25: The wording "rightly or wrongly" suggests a contested issue, and I would suggest re-wording. Is there more background material on tailings ponds and their anaerobic activity that could supplement this discussion?

p. 24 line 28: Describe this methodology, e.g. by changing to "recent core sampling methodology".

Figure 1: Include wind barbs or a wind direction arrow on each map

Figures 2 and 3: The colored markers appear to be wind direction arrows, which is an important parameter in these graphs. However, the arrows are very difficult to see. I recommend mentioning them in the figure captions, and making the markers more

obvious (sparser, outlined, or some other format)

Figure 8: remove "date" from bottom axis

Typos/typesetting

p. 4, line 19: CH4 subscript

p.9 line 17: double-check notation/formatting of U-square. subscripts on sn, s1

p. 12 lines 24-25: degree symbol

p. 24: I suggest more emphasis on Figure 8 in this section (e.g. reference it on line 8)

---

## Author Response (AR1)

Thank-you to both referees for their comments. We respond to both sets of comments here. Black are comments. Our responses are in blue, normal font, with text additions *"also in blue with quotes, italicized and sometimes underlined."* Following is the paper with track changes.

**Anonymous Referee #1

Summary/General comments: Baray et al. present aircraft measurements made around the Athabasca Oil Sands region and employ multiple mass balance approaches to quantify methane emissions from the entire region as well as individual facilities/components to the region. They also use multiple trace gases to attribute to specific processes, and compare results with reported inventorie emissions. This paper is well placed in ACP. This paper contributes to our understanding of methane emissions from a unique but potentially high impact source region. Overall this is a well-written paper, a very nice/sound analysis, and I enthusiastically recommend publishing with only a few minor suggestions.

Response: Thank-you for your efforts. Your comments are appreciated.

Minor comments: Page 4, lines 1-20: this introduction portion is long and dedicated to the recent confusion about global methane and global methane trends. While accurately written, I don't think it is helpful for this paper. Addressing methane emissions from the oil sands is not going to help with these large questions, and motivating the oil sands emissions does not need invoking some the global decadal confusion, but instead could be better motivated focusing on the work in the last 10 years attempting to address methane emissions from the oil and gas production sector, where large discrepancies have been found and this work contributed nicely.

Response: Yes, both reviewers agree on this point. We have now shortened the introduction significantly, while retaining the important points that motivate the current study.

Throughout: Please change the units for methane from ppm to ppb. It is standard to work with methane in ppb, and as the signals observed and discussed make more sense to see in ppb than ppm, this change should be made throughout.

Response: With respect, we would prefer to leave the units presented in many figures and throughout as ppm, rather than ppb. Standardization is not always the best way. At times it is more convenient to use ppm, while sometimes (i.e., when we discuss enhancements) it is more convenient to use ppb. It depends on the size of the numbers. For example, the scale in Fig 2a goes from 2.0 to 4.5 ppm with two significant figures. If converted to ppb, the scale in this figure (and many others) would need to be have four significant figures for every tick, from 2000 to 4500ppm, an unnecessary use of excess figures, in our opinion. At other times when discussing an enhancement of say 1.1 ppm, because of a lack of precision in the number we would be forced to present the enhancement as $1.1 \times 10^3$ ppb if using ppb, so as not to use excessive significant figures. It would be better to discuss such an enhancement as 1.1 ppm. Scientific readers should not have trouble interpreting both ppb and ppm in the same publication we think. We have seen many publications that publish $CH_4$ in ppm, as well as ppb.

Figure 1: Would help a lot to have spatial scale on these figures. Also would be useful to have some wind arrows indicating what winds look like on each of these flight days.

Response: Agreed.  We have now added both a spatial scale and the average wind direction vector for each plot in Figure 1.

Figure 3 (and applies to other plumes): I would like to see what the correlation looks like between different gases within each designated plume. Some tracer-tracer plots with the different plumes shown would be helpful to show/establish how robust the correlations are for each of these tracer-tracer relations.

Response:  We have now included a Figure in Supplemental (see below) that demonstates the correlation of $CH_4$ (vs NOy, rBC, and BTEX), in the major plumes seen in Figure 2.  Text has been added to the main paper alerting the reader to the Supplemental Figure.

*"The in-plume correlations of $CH_4$ with the associated tracers (NOy, rBC and BTEX) for each of the Plumes identified in Figure 2 are shown in Figure S2 (Supplemental Information)."*

*"Fig S2: Correlation Plots for Plumes A-D corresponding to Figure 2 (SML Mine, SML Tailings Pond, SUN Tailings Pond, SUN Mine). $CH_4$ is well correlated with tracer species NOy, BC and BTEX for the various sources. Linear coefficients of determination ($r^2$) are in the range of 0.44-0.83. The lowest $r^2$ values are from the $CH_4$ vs BTEX plot for Plume C and the $CH_4$ vs NOy and $CH_4$ vs BC plots for Plume D. These two sources correspond to lower emissions and mixing ratios of both $CH_4$ and the associated species. In the context of our results, this analysis confirms the correlation of CH4 with various species as shown in Figure 2 which are used to spatially define plume boundaries."*

[Figure]

Page 16, Lines 1-18 as well as Table 1: I'm a little worried about the ethane:methane analysis and would like more supporting information. Smith, Kort, Karion et al., 2015 ES&T used continuous ethane:methane measurements over the Barnett Shale and showed that using limited, discrete flask samples could lead to erroneous ethane:methane ratios. It would help if the authors showed on the time series plot illustrating the plume where in the plume(s) the flasks were collected to help illustrate what the flask ethane may be representative of. The limited discrete samples may have been sufficient, or there may be important gaps causing an uncertainty in how much ethane in fact was emitted – at this point I cannot assess this and this should be improved.

Response: Good point. We have now added a figure in supplemental that shows where the discrete canisters were taken in relation to the $CH_4$ plumes. The key result is that none of the sources mentioned in the paper appear to be significant sources of ethane, and overall there was low ethane observed in the AOSR, consistent with Simpson et al., 2010. We agree that limited discrete sampling can be misleading, and the uncertainty of the ethane/methane ratios will be much higher than would be available if continuous ethane and methane measurements were available. However, our purpose was to demonstrate the low ethane/methane ratios of the sources in the region, in contrast to methane sources in other conventional oil and gas regions. The following text has been added to the paper in this section, including references to Smith et al., 2015. Thanks for pointing out this reference,

*"Source profiles of $CH_4$ are further compared to measurements of ethane ($C_2H_6$). Source-attribution studies for $CH_4$ commonly use higher ethane-to-methane ratios (EMRs) as a signature for oil and gas emissions, on both a regional (Peischl et al., 2016) and global (Hausmann et al., 2016) scale, while low EMR ratios can be indicative of microbial sources of methane that do not emit ethane (agriculture, landfills, wetlands, etc.) (Smith et al., 2015)."*

and

*"The problems associated with determining EMR ratios from a combination of continuous $CH_4$ measurements and discrete canister sampling of ethane from aircraft have been highlighted recently, where it was shown that actual EMR ratios determined in this way can be off by up to a factor of two (Smith et al., 2015). Thus the limited EMR data shown in the Table 1 are not intended to be a comprehensive measure of EMR in the AOSR but simply to support the conclusion that the major sources of methane from the facilities in the AOSR are microbial in nature without a significant co-emission of ethane."*

*"Figure S3: Time series plots of methane (red line) and discrete canisters samples analyzed for ethane (blue lines) corresponding to the same plumes used in Table 1 for the ethane/methane ratio calculations. These are a small subset of the canisters that were sampled over the aircraft campaign. These example plumes attempt to isolate known sources from the three facilities and support the conclusion that there were not any significant sources of ethane in the AOSR, in agreement with Simpson et al., 2010."*

[Figure]

Page 18 Line 6: The vertically varying background can be troublesome/worrisome. It would be helpful to see the profile that is used here and understand how variable the background is.

Response: We have now included the vertical profiles $[CH_4]_B(z)$, for each screen determination, in Figure S1 (Supplemental Information).  The use of vertically variant profiles is an improvement over the use of a single invariant background number at all heights, since on some days, a regional buildup of $CH_4$ from other surface sources upwind of the source in question can occur.  This is an established method in the literature, as stated in Section 2.  We have added the following text in section 2:

*"Example vertical profiles of $[CH_4]_B$ (z) for each day are included in Figure S1 (Supplemental Information)"*

*"Figure S1: Background profiles, $[CH_4]_B$ (z), were selected from regions of the interpolated screens away from plume sources, corresponding to 2-20km spatial lengths depending on the flight paths. Error bars are the 1σ variability within the 2-20km spatial regions of background air. Background $CH_4$ for the vertical regions 150-200m above ground to the surface are estimated based on extrapolations (constant or linear) from the lowest transects to the surface and included in the uncertainty analysis. The lowest aircraft transects usually converged to a constant value (Box 3,5,6,7,9 left to right) or showed a small linear enhancement (Box 2,4,8) which provided best fits to the surface."*

[Figure]

Page26, lines1-2: Should specify the seasonality of fugitive emissions from this unique oil sands source are unknown, not fugitive emissions in general.

Response: Yes. Wording was changed to:

*"…however, we consider this assumption to be highly uncertain as the seasonality of fugitive emissions rates of CH$_4$ in the Athabasca Oil Sands region is still a major uncertainty."*

**Anonymous Referee #2

This paper presents a thorough study of emissions from oil sands facilities. While methane is the focus of these aircraft measurements, a number of complimentary species help to characterize emissions and separate individual sub-sources at each site. The authors do a nice job of contrasting the emissions from the different facilities visited, and bring in previous measurement and inventory work for context. The paper is well-written and organized. The curtain and box methodologies are accurately and simply described. I believe this paper is appropriate for publication in ACP, with only a few minor edits:

Response:  Thank-you for your efforts.  Your comments are appreciated.

The first two and a half pages of introductory material discuss methane and its climate and ozone formation impacts. I think that this background material should be condensed, with more of a focus on the oil sands region.

Response:  Yes, both reviewers agree on this point.  We have now shortened the introduction significantly, while retaining the important points.

The introductory material starting on page 5, line 11 is of utmost interest to this study. I recommend this section be supplemented with a sentence or two about anaerobic methane formation in tailings ponds, which is mentioned briefly later (p. 14 line 5).

Response:  Yes, we have added the following text.

*In particular, a significant fraction of the CH$_4$ is not associated with fossil fuel reserves, but is emitted from the tailings ponds (Small et al., 2015).  The factors giving rise to the release of CH$_4$ from these ponds are complex but include the organic and inorganic chemical composition of the ponds, the diversity and types of microbial communities, especially methanogens, as well as the age of the ponds.  It is reported that it took 20 years and 15 years for the largest ponds at Syncrude and Suncor respectively, to show evidence of methane bubbling from the surface (Small et al., 2015).*

Related to the above comment, on p. 20 line 24, the authors note that younger ponds should produce less methane. The subject of tailing pond methane emissions warrants a paragraph of discussion in the text, explaining why the high emissions from P23 and WIP might be expected (age, any other process differences), and why emissions of methane were low/undetected from other ponds.

Response: Perhaps instead of WIP, you meant MLB? In any case, a few lines will be added, yes. A paragraph of discussion on this topic might be beyond the scope of this paper though and we want to avoid being speculative, going beyond the scope of the paper. A few lines were added as such…

*"Our method requires $CH_4$ plumes to be clearly enhanced above background, so trace amounts of $CH_4$ from ponds in the other three facilities were not detected. This could be related to differences in the chemical composition of the process streams being released into these ponds, or it could simply be due to these ponds being younger in age, with insufficient time for the anaerobic methanogenic communities to be established (Small et al., 2015)."*

p.14, line 19: "we did not detect methane": Were any canisters taken showing light hydrocarbon enhancements?

Response: No and yes. A few canisters were taken on these flights, but unfortunately they were too sparse and uncertain as to isolate the tailings ponds emissions. Canister samples were opportunistic, usually taken downwind of facilities within major plumes identified by the major pollutants (NOx, SOx, $CH_4$, $CO_2$). Perhaps future studies could address this.

p. 20 line 12: Is there a reference or previous study that looks at this degassing rate?

Response: We are not aware of such a study. However, the stated assumption is scientifically reasonable. The only analogy we can draw is to the difference in vapor composition of a spilled gasoline sample vs gasoline vapor in equilibrium with liquid gasoline. In the former case, due to the time allowed for complete evaporation (minutes to hours depending on the size of the spilled gasoline sample), the vapor composition is identical to the spilled sample composition...quantitatively, due to mass balance (This is what we expect for methane). For the equilibrated vapor sample, the vapor composition reflects the distilled vapor composition which can be predicted using the saturated vapor pressure of the individual components, their mole ratios, and Raoult's Law. Methane is more volatile than any individual component of a gasoline sample. We have reworded the sentence to say…

*"This is reasonable considering that it would be expected that degassing of an extremely volatile gas such as $CH_4$ from the oil sands material would be quantitative in a short period of time after the ore is exposed or crushed."*

p. 22 line 19, and elsewhere: When discussing seasonality of emissions, it would be useful to remind the reader that these measurements were taken in August-September

Response: We agree such a reminder would be useful here. For most effect, we add the reminder in the very last sentence, before moving to Section 3.5 on comparison to emission inventories…

*"The two values are combined here using an error-weighted uncertainty resulting in a final AOSR facility emissions estimate of 19.6 ± 1.1 tonnes $CH_4$ $hr^{-1}$, measured during a summertime period."*

p. 23, line 25: The wording "rightly or wrongly" suggests a contested issue, and I would suggest re-wording. Is there more background material on tailings ponds and their anaerobic activity that could supplement this discussion?

Response: " rightly or wrongly" has been removed with a slight rewording of the sentence to ….

*"The argument used to justify the use of a constant seasonal temporal factor in the GOA report is that temperatures at depth in a tailings pond are said to remain relatively constant throughout the year and thus, biogenic gas formation continues in the winter (GOA, 2014)."*

p. 24 line 28: Describe this methodology, e.g. by changing to "recent core sampling methodology".

Response: Yes. As suggested this has been changed to *"…recent core sampling methodology…"*

Figure 1: Include wind barbs or a wind direction arrow on each map

Response: We have now added both a spatial scale and the average wind direction vector for each plot.

Figures 2 and 3: The colored markers appear to be wind direction arrows, which is an important parameter in these graphs. However, the arrows are very difficult to see. I recommend mentioning them in the figure captions, and making the markers more obvious (sparser, outlined, or some other format).

Response: We have now mentioned them in the figure caption, making them sparser would mean removing data points as well. We feel it may be too sparse. The large red arrows in the figure are a backwards extension of the small wind barbs, which give a good idea of wind direction.

Text has been added to each Figure caption in Fig 2 and Fig 3…

*"Each data point is color coded for $CH_4$ mixing ratio as well as instantaneous wind vector measured on the aircraft at that location."*

Figure 8: remove "date" from bottom axis

Response Yes, the label (including date) has now been removed from the Figure.

**Typos/typesetting**

p. 4, line 19: CH4 subscript

Response:  fixed.

p.9 line 17: double-check notation/formatting of U-square. subscripts on sn, s1

Response: fixed.

p. 12 lines 24-25: degree symbol

Response: fixed.

p. 24: I suggest more emphasis on Figure 8 in this section (e.g. reference it on line 8)

Response: Our reference to Fig 8 in the text is somewhat later, yes. We have now added the reference sooner in the section…

*"The annual emission rates of $CH_4$ extracted from the inventory were downscaled to hourly emissions rates for comparison with our measurements with an assumption of equal seasonal and diurnal profiles 365 days a year, 24 hours per day; for consistency with upscaling factors used to generate annual emissions **(see Figure 8)**."*

Most of the section before this is a discussion of how we get the last column in Fig 8…Inventory.

We then have a full paragraph discussing Fig 8, which we feel is perhaps adequate.

[revised manuscript text omitted]

S. Baray[1], A. Darlington[2], M. Gordon[3], K.L. Hayden[2], Amy Leithead[2], S.-M. Li[2], P.S.K. Liu[2], R.L. Mittermeier[2], S.G. Moussa[2], J. O'Brien[2], R. Staebler[2], M. Wolde[4], D. Worthy[5], R. McLaren[1]

1. Centre for Atmospheric Chemistry, York University, Toronto 2. Air Quality Research Division, Environment and Climate Change Canada, Toronto 3. Earth and Space Science and Engineering, York University, Toronto 4. National Research Council of Canada, Ottawa 5. Climate Research Division, Environment and Climate Change Canada, Toronto, Canada

Corresponding authors: Robert McLaren (rmclaren@yorku.ca) & Katherine Hayden (katherine.hayden@canada.ca)

**Assessment of Uncertainties**

Tables S1-6 show the results of the sensitivity analysis to estimate contributions to total uncertainty. Parameters contributing to uncertainties depend on the mass balance method used and the screen-based (Eq. 1) or the box-approach (Eq. 2). Minor uncertainties that contribute to both methods are errors in the $CH_4$ mixing ratio measurement and wind measurements. $CH_4$ measurement errors from the instrument are <1%. Measurements of trace species from other instruments were used qualitatively to deduce plume origins, thus they do not contribute to total uncertainties. In a previous study, a Monte Carlo simulation was used to demonstrate the wind measurements contribute <1% to the change in uncertainties (Gordon et al., 2016). A significant source of uncertainty for both mass balance methods is the extrapolation of $CH_4$ mixing ratios to the surface for ground-level plumes. Surface extrapolation uncertainties are highly variable with flight, consistent with the literature. Cambaliza et al. (2014) found surface extrapolation uncertainties to be 4, 9 and 16% for three different mass balance flights downwind of Indianapolis to determine $CH_4$ fluxes, and Gordon et al., 2016 found this to be 15% and 26% for two Oil Sands flights for the CNRL facility. The uncertainty depends on the range of surface mixing ratios resulting from fitting varying extrapolation methods. We derive a range of possible emissions rates by comparing results from constant, linear and half-Gaussian extrapolations to the surface. $CH_4$ measurements at Fort McKay are used as constraints on surface mixing ratios when flight paths are

directly overhead (Aug 16 Flight 4A, SML and SUN). Half-guassian extrapolations are used where fits are above constraints ($r^2>0.40$). Future studies can further minimize these uncertainties with simultaneous ground-level mixing ratio measurements.

5    Additional uncertainties specific to the box-approach (Eq. 2) are assessed according to the methodology described in Gordon et al., 2016. Contributing factors are: (1) the uncertainty in the box-top height (affecting the $E_{CH}$ and $E_{CV}$ terms), estimated by reducing the box height by 100 m, (2) changes in air mass density within the volume of the box (affecting $E_{CM}$), estimated using the minimum and maximum of pressure and temperature ratios derived from surrounding meteorological stations, (3) inclusion of

10    the estimated vertical turbulence term ($E_{CVT}$), and (4) uncertainty in the mean $CH_4$ mixing ratio at the box-top (affecting $E_{CV}$) determined from the 95% confidence interval ($2\sigma/\sqrt{n}$) of interpolated measurements. These terms are recalculated according to the range of possible input parameters in order to derive resulting uncertainties in the emissions rates. Screen-approach specific uncertainties (Eq. 1) are mostly due to the variability in the background mixing ratio $[CH_4]_B$, determined using the outer

15    edges of the screen away from plume sources (screen flights) and upwind measurements (box flights). For each flight measurements from multiple background regions (>1km) occurring closely in time are used as possible inputs, which are identified clearly due to the high $CH_4$ mixing ratios observed from plumes. Other sources of uncertainty are the vertical extent of the screen (upper bound, z) and the horizontal boundaries ($s_1$-$s_2$) of individually characterized plumes. These plume boundaries are

20    expanded and contracted to derive a range of possible integrals.

Uncertainties for each mass balance flight are added in quadrature to derive a range of possible emissions rates. Estimates for the same source category within a facility, as well as total estimates for the same facility, are treated as independent estimates and combined using an error-weighted mean

25    ($1/\sigma^2$).

**Meteorological Conditions**

Tables S1-6 (bottom) present various flight details and meteorology. Flights used are those with a high number of aircraft transects (≥6) that show full characterization of plume vertical extent. Boundary layer heights are determined using visual inspection of dew point temperature alongside LIDAR backscatter reports from ground-site AMS13 during flight times. Ground temperature and wind direction measurements are based on ground-site data at AMS13 over the course of the day. Wind speeds shown are from interpolated screens $\pm 1\sigma$.

Table S1-6: **Top: Sensitivity analysis displaying uncertainty contributions ($1\sigma$) shown in percent change from the best-estimate emissions rate, added in quadrature for totals. Uncertainties in individual plumes are noted with superscripts for tailings ponds (t), mines (m) and facility/other (f). Screen estimates using an overlapping subset of downwind measurements from a box flight of the same day are shown with an asterisk (\*). Middle: List of emissions rates for source categories and facility totals in tonnes $CH_4$ per hour (tonnes $hr^{-1}$). Bottom: Various aircraft flight details and meteorological parameters.**

**Table S1: Syncrude Mildred Lake (SML)**

| | | Aug 14 Box | Aug 14 Screen A* | Aug 14 Screen B | Aug 16 Screen A |
|---|---|---|---|---|---|
| | Measurement Error (%) | 1 | 1 | 1 | 1 |
| | Wind Error (%) | 1 | 1 | 1 | 1 |
| | Surface Extrapolation (%) | 4 | 11 | 3 | 28 |
| Box | Box-top Height (%) | 15 | | | |
| | Density Change (%) | 11 | | | |
| | Vertical Turbulence (%) | 2 | | | |
| | Box-Top Mixing Ratio (%) | **4** | | | |
| Screen | Background Mixing Ratio (%) | | 13 | 19 | 8 |
| | Screen-Top Height (%) | | 6 | 6 | 1 |
| | Plume Separation (%) | | 6$^t$, 11$^m$ | 5$^t$, 12$^m$ | 5$^t$, 8$^m$ |
| | Total Uncertainty Facility (%) | 20 | 19 | 21 | 30 |
| | Total Uncertainty Plumes (%) | | 20$^t$, 22$^m$ | 21$^t$, 24$^m$ | 30$^t$, 31$^m$ |
| | Emissions Rate Ponds (tonnes hr$^{-1}$) | | 6.38 ± 1.23 | 5.83 ± 1.22 | 8.63 ± 2.59 |
| | Emissions Rate Mines (tonnes hr$^{-1}$) | | 2.71 ± 0.60 | 2.67 ± 0.64 | 3.07 ± 0.95 |
| | Emissions Rate Facility/Other (tonnes hr$^{-1}$) | | | | |
| | **Emissions Rate Total (tonnes hr$^{-1}$)** | **7.68 ± 1.54** | **9.10 ± 1.73** | **8.50 ± 1.79** | **11.82 ± 3.55** |
| | Aircraft Transect Count | 6 | 6 | 8 | 9 |
| | Boundary Layer Height (m agl) | 360-400 | 360-400 | 400-600 | 350-400 |
| | Temperature (°C) | 20.8 ± 6.0 | 20.8 ± 6.0 | 20.8 ± 6.0 | 19.5 ± 3.8 |
| | Wind Speed (m/s) | 3.1 ± 2.5 | 3.1 ± 2.5 | 5.1 ± 1.6 | 2.8 ± 0.8 |
| | Daily Mean Wind Direction (°) | 220 ± 37 | 220 ± 37 | 220 ± 37 | 225 ± 57 |

**Table S2: Suncor Energy OSG (SUN)**

|  |  | Aug 16 Screen A | Aug 29 Box | Aug 29 Screen* |
|---|---|---|---|---|
|  | Measurement Error (%) | 1 | 1 | 1 |
|  | Wind Error (%) | 1 | 1 | 1 |
|  | Surface Extrapolation (%) | 4 | 14 | 4 |
| Box | Box-top Height (%) |  | 1 |  |
|  | Density Change (%) |  | 17 |  |
|  | Vertical Turbulence (%) |  | 2 |  |
|  | Box-Top Mixing Ratio (%) |  | 5 |  |
| Screen | Background Mixing Ratio (%) | 23 |  | 2 |
|  | Screen-Top Height (%) | 1 |  | 9 |
|  | Plume Separation (%) | $12^t$, $1^m$ |  | $9^t$, $9^m$ |
|  | Total Uncertainty Facility (%) | 24 | 23 | 11 |
|  | Total Uncertainty Plumes (%) | $27^t$, $24^m$ |  | $14^t$, $14^m$ |
|  | Emissions Rate Ponds (tonnes hr$^{-1}$) | $3.16 \pm 0.85$ |  | $2.30 \pm 0.32$ |
|  | Emissions Rate Mines (tonnes hr$^{-1}$) | $1.53 \pm 0.37$ |  | $1.88 \pm 0.26$ |
|  | Emissions Rate Facility/Other (tonnes hr$^{-1}$) |  |  |  |
|  | **Emissions Rate Total (tonnes hr$^{-1}$)** | **$4.69 \pm 1.13$** | **$3.96 \pm 0.91$** | **$4.18 \pm 0.42$** |
|  | Aircraft Transect Count | 9 | 7 | 7 |
|  | Boundary Layer Height (m agl) | 350-400 | 400-500 | 400-500 |
|  | Temperature (°C) | $19.5 \pm 3.8$ | $15.2 \pm 2.4$ | $15.2 \pm 2.4$ |
|  | Wind Speed (m/s) | $2.8 \pm 0.8$ | $1.8 \pm 1.3$ | $1.8 \pm 1.3$ |
|  | Daily Mean Wind Direction (°) | $225 \pm 57$ | $26 \pm 40$ | $26 \pm 40$ |

**Table S3: Canadian National Resources Limited Horizon (CNRL)**

| | | Aug 20 Box | Aug 20 Screen* | Sep 02 Box | Sep 02 Screen* |
|---|---|---|---|---|---|
| | Measurement Error (%) | 1 | 1 | 1 | 1 |
| | Wind Error (%) | 1 | 1 | 1 | 1 |
| | Surface Extrapolation (%) | 22 | 26 | 12 | 11 |
| Box | Box-top Height (%) | 1 | | 18 | |
| | Density Change (%) | 5 | | 6 | |
| | Vertical Turbulence (%) | 2 | | 7 | |
| | Box-Top Mixing Ratio (%) | 3 | | 8 | |
| Screen | Background Mixing Ratio (%) | | 16 | | 25 |
| | Screen-Top Height (%) | | 5 | | 2 |
| | Plume Separation (%) | | | | $6^m$, $12^f$ |
| | Total Uncertainty Facility (%) | 23 | 31 | 25 | 28 |
| | Total Uncertainty Plumes (%) | | | | $29^m$, $30^f$ |
| | Emissions Rate Ponds (tonnes hr$^{-1}$) | | | | |
| | Emissions Rate Mines (tonnes hr$^{-1}$) | | | | $2.56 \pm 0.74$ |
| | Emissions Rate Facility/Other (tonnes hr$^{-1}$) | | | | $0.98 \pm 0.29$ |
| | **Emissions Rate Total (tonnes hr$^{-1}$)** | **3.65 ± 0.84** | **3.67 ± 1.14** | **3.53 ± 0.88** | **3.54 ± 1.00** |
| | Aircraft Transect Count | 12 | 12 | 10 | 10 |
| | Boundary Layer Height (m agl) | 700-900 | 700-900 | 600-1000 | 600-1000 |
| | Temperature (°C) | 16.3 ± 4.3 | 16.3 ± 4.3 | 12.7 ± 5.1 | 12.7 ± 5.1 |
| | Wind Speed (m/s) | 2.4 ± 1.9 | 2.4 ± 1.9 | 5.9 ± 2.8 | 5.9 ± 2.8 |
| | Daily Mean Wind Direction (°) | 262 ± 35 | 262 ± 35 | 338 ± 59 | 338 ± 59 |

**Table S4: Shell Albian and Jackpine (SAJ)**

| | | Aug 21 Box | Aug 21 Screen* | Sep 06 Box | Sep 06 Screen* |
|---|---|---|---|---|---|
| | Measurement Error (%) | 1 | 1 | 1 | 1 |
| | Wind Error (%) | 1 | 1 | 1 | 1 |
| | Surface Extrapolation (%) | 5 | 7 | 12 | 7 |
| Box | Box-top Height (%) | 8 | | 5 | |
| | Density Change (%) | 10 | | 16 | |
| | Vertical Turbulence (%) | 5 | | 2 | |
| | Box-Top Mixing Ratio (%) | 9 | | 7 | |
| Screen | Background Mixing Ratio (%) | | 27 | | 17 |
| | Screen-Top Height (%) | | 10 | | 5 |
| | Plume Separation (%) | | | | |
| | Total Uncertainty Facility (%) | 18 | 30 | 22 | 20 |
| | Total Uncertainty Plumes (%) | | | | |
| | Emissions Rate Ponds (tonnes hr$^{-1}$) | | | | |
| | Emissions Rate Mines (tonnes hr$^{-1}$) | | 1.44 ± 0.43 | | 1.18 ± 0.24 |
| | Emissions Rate Facility/Other (tonnes hr$^{-1}$) | | | | |
| | **Emissions Rate Total (tonnes hr$^{-1}$)** | **1.60 ± 0.29** | **1.44 ± 0.43** | **1.25 ± 0.28** | **1.18 ± 0.24** |
| | Aircraft Transect Count | 10 | 10 | 10 | 10 |
| | Boundary Layer Height (m agl) | 1200-1500 | 1200-1500 | 900-1200 | 900-1200 |
| | Temperature (°C) | 16.5 ± 3.6 | 16.5 ± 3.6 | 14.8 ± 6.2 | 14.8 ± 6.2 |
| | Wind Speed (m/s) | 1.3 ± 0.8 | 1.3 ± 0.8 | 4.3 ± 0.9 | 4.3 ± 0.9 |
| | Daily Mean Wind Direction (°) | 258 ± 50 | 258 ± 50 | 7 ± 50 | 7 ± 50 |

**Table S5: Syncrude Aurora (SAU)**

| | | Aug 29 Box | Aug 29 Screen* | Sep 06 Screen* |
|---|---|---|---|---|
| | Measurement Error (%) | 1 | 1 | 1 |
| | Wind Error (%) | 1 | 1 | 1 |
| | Surface Extrapolation (%) | 10 | 14 | 6 |
| Box | Box-top Height (%) | 4 | | |
| | Density Change (%) | 9 | | |
| | Vertical Turbulence (%) | 2 | | |
| | Box-Top Mixing Ratio (%) | 3 | | |
| Screen | Background Mixing Ratio (%) | | 11 | 13 |
| | Screen-Top Height (%) | | 4 | 13 |
| | Plume Separation (%) | | | |
| | Total Uncertainty Facility (%) | 15 | 19 | 20 |
| | Total Uncertainty Plumes (%) | | | |
| | Emissions Rate Ponds (tonnes hr$^{-1}$) | | | |
| | Emissions Rate Mines (tonnes hr$^{-1}$) | | 1.29 ± 0.25 | 1.56 ± 0.31 |
| | Emissions Rate Facility/Other (tonnes hr$^{-1}$) | | | |
| | **Emissions Rate Total** (tonnes hr$^{-1}$) | **1.70 ± 0.26** | **1.29 ± 0.25** | **1.56 ± 0.31** |
| | Aircraft Transect Count | 3 | 3 | 10 |
| | Boundary Layer Height (m agl) | 400-500 | 400-500 | 900-1200 |
| | Temperature (°C) | 15.2 ± 2.4 | 15.2 ± 2.4 | 14.8 ± 6.2 |
| | Wind Speed (m/s) | 2.3 ± 0.7 | 2.3 ± 0.7 | 4.3 ± 0.9 |
| | Daily Mean Wind Direction (°) | 26 ± 40 | 26 ± 40 | 7 ± 50 |

**Table S6: Total Oil Sands Screen**

|  |  | Aug 16 Screen B |
|---|---|---|
|  | Measurement Error (%) | 1 |
|  | Wind Error (%) | 1 |
|  | Surface Extrapolation (%) | 3 |
| Box | Box-top Height (%) |  |
|  | Density Change (%) |  |
|  | Vertical Turbulence (%) |  |
|  | Box-Top Mixing Ratio (%) |  |
| Screen | Background Mixing Ratio (%) | 14 |
|  | Screen-Top Height (%) | 5 |
|  | Plume Separation (%) |  |
|  | Total Uncertainty Facility (%) | 16 |
|  | Total Uncertainty Plumes (%) |  |
|  | Emissions Rate Ponds (tonnes hr$^{-1}$) |  |
|  | Emissions Rate Mines (tonnes hr$^{-1}$) |  |
|  | Emissions Rate Facility/Other (tonnes hr$^{-1}$) |  |
|  | **Emissions Rate Total** (tonnes hr$^{-1}$) | **23.6± 3.8** |
|  | Aircraft Transect Count | 10 |
|  | Boundary Layer Height (m agl) | 400-450 |
|  | Temperature (°C) | 19.5 ± 3.8 |
|  | Wind Speed (m/s) | 2.8 ± 1.0 |
|  | Daily Mean Wind Direction (°) | 225 ± 57 |

Figure S1: Background profiles, [CH4]B(z), were selected from regions of the interpolated screens away from plume sources, corresponding to 2-20km spatial lengths depending on the flight paths. Error bars are the 1σ variability within the 2-20km spatial regions of background air. Background CH4 for the vertical regions 150-200m above ground to the surface are estimated based on extrapolations (constant or linear) from the lowest transects to the surface and included in the uncertainty analysis. The lowest aircraft transects usually converged to a constant value (Box 3,5,6,7,9 left to right) or showed a small linear enhancement (Box 2,4,8) which provided best fits to the surface.

[Figure]

Figure S2: Correlation Plots for Plumes A-D corresponding to Figure 2 (SML Mine, SML Tailings Pond, SUN Tailings Pond, SUN Mine). $CH_4$ is well correlatred with tracer species $NO_y$, BC and BTEX for the various sources. Linear coefficients of determination ($r^2$) are in the range of 0.44-0.83. The lowest $r^2$ values are from the $CH_4$ vs BTEX plot for Plume C and the $CH_4$ vs $NO_y$ and $CH_4$ vs BC plots for Plume D. These two sources correspond to lower emissions and mixing ratios of both $CH_4$ and the associated species. In the context of our results, this analysis confirms the correlation of $CH_4$ with various species as shown in Figure 2 which are used to spatially define plume boundaries.

[Figure]

Figure S3: Time series plots of methane (red line) and discrete canisters samples analyzed for ethane (blue lines) corresponding to the same plumes used in Table 1 for the ethane/methane ratio calculations. These are a small subset of the canisters that were sampled over the aircraft campaign. These example plumes attempt to isolate known sources from the three facilities and support the conclusion that there were not any significant sources of ethane in the AOSR, in agreement with Simpson et al., 2010.

[Figure]

---

## Author Response (AR2)

Thanks to the co-editor for her work on this manuscript.  Only one change was optionally requested (a matter of personal preference), a change of all the units for methane from ppm to ppb.  As the editor said this was optional, we have taken the option of not making the change on 7 figures; we prefer to retain the ppm units.

Thanks

Rob McLaren